# How well are aerosol-cloud interactions represented in climate models? – Part 2: Isolating the aerosol impact on clouds following the 2014–15 Holuhraun eruption

George Jordan<sup>1</sup>, Florent Malavelle<sup>2</sup>, Jim Haywood<sup>1,3</sup>, Ying Chen<sup>4</sup>, Ben Johnson<sup>1</sup>, Daniel Partridge<sup>3</sup>, Amy Peace<sup>1</sup>, Eliza Duncan<sup>3</sup>, Duncan Watson-Parris<sup>5</sup>, David Neubauer<sup>6</sup>, Anton Laakso<sup>7</sup>, Martine Michou<sup>8</sup>, and Pierre Nabat<sup>8</sup>

**Correspondence:** George Jordan (george.jordan@metoffice.gov.uk)

# Abstract.

Aerosols significantly influence Earth's radiative balance, yet considerable uncertainty exists in the underpinning mechanisms, particularly those involving clouds, Aerosol-cloud interactions (ACIs) are the most uncertain element in anthropogenic radiative forcing, hampering our ability to constrain Earth's climate sensitivity and understand future climate change. The 2014–2015 Holuhraun volcanic eruption in Iceland released sulphur dioxide (SO<sub>2</sub>) into the lower troposphere on a level comparable to continental-scale emissions. The resultant volcanic plume across an often near-pristine region of the northern North Atlantic Ocean presents an ideal opportunistic experiment to explore ACI representation within general circulation models (GCMs). We present Part 2 of a two-part AeroCom (Aerosol Comparisons between Observations and Models) Phase III intermodel comparison study that utilises satellite remote sensing observations to assess modelled cloud responses to the volcanic aerosol within 8 state-of-the-art GCMs during September and October 2014. We isolate the aerosol effect from meteorological variability and find that the GCMs – particularly their multi-model ensemble response – adeptly capture the observed cloud microphysical changes associated with the ACI first indirect effect (i.e., Twomey effect). Meanwhile, a clear divergence exists in the GCM responses of large-scale cloud properties, namely cloud liquid water content, expected from the precipitation suppression mechanism of the ACI second indirect effect (i.e., rapid adjustments). We attribute this to limitations and differences in their autoconversion schemes under high aerosol loading, specifically in sub-grid variability representations. Finally, our multi-model ensemble estimates that Holuhraun had a global radiative forcing of  $-0.11 \pm 0.04~\mathrm{Wm^{-2}}$  across September and October 2014.

<sup>&</sup>lt;sup>1</sup>Met Office Hadley Centre, Exeter, UK

<sup>&</sup>lt;sup>2</sup>Met Office, Exeter, UK

<sup>&</sup>lt;sup>3</sup>Department of Mathematics and Statistics, University of Exeter, Exeter, UK

<sup>&</sup>lt;sup>4</sup>School of Geography, Earth and Environmental Sciences, University of Birmingham, Birmingham, UK

<sup>&</sup>lt;sup>5</sup>Scripps Institution of Oceanography and Halıcıoğlu Data Science Institute, University of California San Diego, California, USA

<sup>&</sup>lt;sup>6</sup>Institute for Climate and Atmospheric Science, ETH Zurich, Zurich, Switzerland

<sup>&</sup>lt;sup>7</sup>Atmospheric Research Centre of Eastern Finland, Finnish Meteorological Institute, Kuopio, Finland

<sup>&</sup>lt;sup>8</sup>CNRM, Université de Toulouse, Météo-France, CNRS, Toulouse, France

#### 1 Introduction

Aerosols have a major influence on the Earth's energy budget through their interactions with solar and terrestrial radiation via direct and indirect mechanisms. The direct mechanism — termed aerosol-radiation interactions — describes the scattering and absorption of radiation by the aerosol itself (e.g., Bellouin et al., 2020; Myhre et al., 2013), whilst the indirect mechanism — known as aerosol-cloud interactions (ACIs) — centres on changes to cloud properties caused by aerosols via their role as cloud condensation nuclei (CCN) (e.g., Bellouin et al., 2020; Fan et al., 2016). Overall, aerosols exert a negative radiative forcing (RF) on the Earth helping offset a portion of the warming from increased greenhouse gas emissions, yet the magnitude of this key effect continues to be a major source of uncertainty in anthropogenic climate change (Forster et al., 2021; Gryspeerdt et al., 2020; Watson-Parris and Smith, 2022). This uncertainty stems predominantly from ACIs, meaning it is of paramount importance that we improve our knowledge of these cloud-mediated processes to improve future climate estimates.

Aerosols prompt cloud modifications through a causal network of events (e.g., Haywood and Boucher, 2000; Fan et al., 2016). For liquid-only clouds, added aerosol can serve as additional CCN which increases cloud droplet number concentration  $(N_d)$  (Twomey, 1974). Holding cloud liquid water content constant (cloud liquid water path, LWP), an increase in  $N_d$  leads to a decrease in cloud droplet size (cloud droplet effective radius,  $r_e$ ), causing an enhancement in cloud albedo (Twomey, 1977). This chain of events is referred to as the "first indirect effect" or the "Twomey effect". Furthermore, smaller cloud droplets decrease the efficiency of collision-coalescence processes delaying the formation of precipitation. Consequently, liquid clouds polluted by aerosol may have longer lifetimes and/or greater cloud fraction (CF) (Albrecht, 1989), and increased depth (Pincus and Baker, 1994), all of which act to increase LWP and further enhance cloud albedo. This subsequent chain of events has historically been referred to as the "second indirect effect", although now further aerosol-induced cloud adjustments are often captured under this term too. Such adjustments include those in non-precipitating clouds whereby the aerosol-induced reduction in  $r_e$  increases evaporation and decreases sedimentation, causing feedbacks that help accelerate entrainment and deplete LWP (Ackerman et al., 2004; Bretherton et al., 2007; Hill et al., 2009; Small et al., 2009). For mixed-phase and ice-only clouds, additional cloud modification processes exist (e.g., Bellouin et al., 2020; Fan et al., 2016; Forster et al., 2021). The myriad of mechanisms underpinning ACIs — each with their own dependency on conditions both meteorological (e.g., atmospheric stability, humidity, temperature) and environmental (e.g., aerosol background concentrations, marine versus land region) — is testament to how challenging constraining ACI uncertainty is.

To alleviate this complexity, studies can focus on aerosol perturbations to systems where the meteorology and environment are well understood. Known as "opportunistic experiments", these instances include industrial plumes, ship tracks, wildfires, regulatory changes, and volcanic eruptions (Christensen et al., 2022). A notable example of the latter is the Holuhraun eruption; an effusive eruption that occurred continuously between the 31<sup>st</sup> August 2014 and 27<sup>th</sup> February 2015 in the Bárðarbunga volcanic system in Iceland (64.85 °N, 16.83 °W) (Gislason et al., 2015; Pedersen et al., 2017). Characterised by its non-explosive nature, Holuhraun released an estimated 9.6–11.8 Tg of sulphur dioxide (SO<sub>2</sub>) (Gislason et al., 2015; Pfeffer et al., 2018) — approximately one-tenth of current global annual anthropogenic SO<sub>2</sub> emissions (Aas et al., 2019; Szopa et al., 2021) — into the lower troposphere (Carboni et al., 2019; Flower and Kahn, 2020; Pfeffer et al., 2018). These SO<sub>2</sub> emissions

subsequently oxidised to sulphate aerosol ( $SO_4^{2-}$ ) leading to the formation of a vast aerosol plume. Such widespread pollution to an often near-pristine region of the northern North Atlantic Ocean over a 6-month duration has made Holuhraun a focal point in studying ACIs at the climatic scale.

55

70

Previous Holuhraun studies have provided valuable insight into ACIs through a variety of approaches. For example, Malavelle et al. (2017) and Zoëga et al. (2023) use general circulation models (GCMs) to generate climatologies within the North Atlantic Ocean and Arctic Ocean respectively, enabling the volcanic aerosol effect on cloud properties to be disentangled from meteorological variability. Both studies find that GCMs simulate a decrease in  $r_e$  during the months following the eruption, yet their LWP responses range from negligible change to a strong increase. Alternatively, Haghighatnasab et al. (2022) and Peace et al. (2024) use an "in-plume versus out-of-plume" approach to isolate the aerosol-induced cloud impacts during September 2014. The studies find increases in  $N_d$  and decreases in  $r_e$  inside the plume compared to outside, whereas the in-plume changes to LWP are mixed and hard to isolate. Moreover, Zoëga et al. (2025) use a GCM to explore the cloud response sensitivity to Holuhraun with respect to eruption season and size of emissions, noting a stronger response occurs during Spring and Summer, and a plateauing of the response with increasing emissions. McCoy and Hartmann (2015) perform an entirely observational based study, noting a decrease in  $r_e$  post-eruption, yet no appreciable changes in LWP or CF. Additionally, Chen et al. (2022) trained a machine learning model to produce a "counterfactual" satellite remote sensing representation of the region absent of Holuhraun emissions, again finding that  $N_d$  increases and  $r_e$  decreases due to the eruption, with minimal changes to LWP. Interestingly, Chen et al. (2022) propose that the additional aerosol prompted a 10 % increase in cloud cover; a result not found in other Holuhraun studies exploring this cloud property.

Here we build on this established set of works by presenting Part 2 of a two-part AeroCom (Aerosol Comparisons between Observations and Models) Phase III inter-model comparison study of the Holuhraun plume and its interactions with clouds. In Part 1, the spatial and chemical evolution of the volcanic plume was assessed (Jordan et al., 2024). Differences in the secondary  $SO_4^{2-}$  aerosol production amongst the GCMs, as well as with observations, were noted, yet overall the modelled representations of the Holuhraun plume were deemed sufficient to explore the impacts of the eruption on ACIs in the region. Here we follow on from Part 1 and assess the ACI representations from 8 state-of-the-art GCMs against satellite remote sensing observations. Here we focus on stratocumulus clouds over an often near-pristine marine region (i.e., minimal anthropogenic influence). We narrow our attention to September and October 2014 as these months offer the most favourable conditions for isolating the aerosol signal relative to the later months of the eruption. This is due to: a) more reliable satellite retrievals, b) peak  $SO_2$  emissions, c) a well-defined volcanic plume with minimal plume dilution, and d) a plethora of studies providing insights on the conditions of this period. We compare model analyses and observations to identify differences in ACI representations, seeking to understand the point at which the models depart from the observed ACI casual chain. We conclude with an updated multi-model ensemble forcing estimate of the Holuhraun eruption.

**Table 1.** Models used in this study. Aerosol module: name of the aerosol module with type given in brackets. Stratiform cloud microphysics: name of the stratiform cloud microphysics scheme (MG1.5 – Gettelman and Morrison (2015); Morrison and Gettelman (2008); Lopez – Lopez (2002); Lohmann – Lohmann et al. (2007); Lohmann and Hoose (2009); Lohmann and Neubauer (2018); P3 – Dietlicher et al. (2018); WB –Wilson and Ballard (1999)). Activation: name of the cloud droplet activation scheme (ARG – Abdul-Razzak and Ghan (2000); Menon – Menon et al. (2002)). Autoconversion: name of the autoconversion parametrisation (KK – Khairoutdinov and Kogan (2000); Smith – Smith (1990)). ACIs: aerosol indirect effects represented. Lat. x long.: atmospheric grid resolution. Levs.: number of vertical levels. References: key references.

| Model name<br>(Full name if applicable)             | Aerosol module<br>(Type)                  | Strat. cloud<br>microphysics | Activation | Auto-<br>conversion | ACIs                  | Lat. x long.    | Levs. | References                                                           |
|-----------------------------------------------------|-------------------------------------------|------------------------------|------------|---------------------|-----------------------|-----------------|-------|----------------------------------------------------------------------|
| CAM5.3-Oslo                                         | OsloAero5.3<br>(Prodtagged <sup>1</sup> ) | MG1.5                        | ARG        | KK                  | Both                  | 0.9° x 1.25°    | 30    | Kirkevåg et al. (2018);<br>Liu et al. (2016);<br>Neale et al. (2012) |
| CNRM-ESM2-1                                         | TACTICv2<br>(Sectional)                   | Lopez                        | Menon      | Smith               | First <sup>2, 3</sup> | 1.41° x 1.41°   | 91    | Michou et al. (2015, 2020<br>Séférian et al. (2019)                  |
| ECHAM6-HAM<br>(ECHAM6.3-HAM2.3)                     | HAM-M7<br>(Modal)                         | Lohmann                      | ARG        | KK                  | Both                  | 1.875° x 1.875° | 47    | Neubauer et al. (2019)<br>Tegen et al. (2019)                        |
| ECHAM6-HAM-P3<br>(ECHAM6.3<br>-HAM2.3-P3)           | HAM-M7<br>(Modal)                         | Р3                           | ARG        | KK                  | Both                  | 1.875° x 1.875° | 47    | Dietlicher et al. (2018)                                             |
| ECHAM6-SALSA<br>(ECHAM6.3<br>-HAM2.3-SALSA)         | HAM-SALSA<br>(Sectional)                  | Lohmann                      | ARG        | KK                  | Both                  | 1.875° x 1.875° | 47    | Kokkola et al. (2018)                                                |
| HadGEM3<br>(HadGEM3-GA7.0)                          | GLOMAP-mode<br>(Modal)                    | $\mathrm{WB}^4$              | ARG        | KK                  | Both <sup>3</sup>     | 1.875° x 1.25°  | 85    | Walters et al. (2019)                                                |
| UKEMS1<br>(UKESM1.0; Boundary<br>Nucleation Off)    | GLOMAP-mode<br>(Modal)                    | $\mathrm{WB}^4$              | ARG        | KK                  | Both <sup>3</sup>     | 1.875° x 1.25°  | 85    | Mulcahy et al. (2020)                                                |
| UKESM1-BLN<br>(UKESM1.0; Boundary<br>Nucleation On) | GLOMAP-mode<br>(Modal)                    | $\mathrm{WB}^4$              | ARG        | KK                  | Both <sup>3</sup>     | 1.875° x 1.25°  | 85    | Mulcahy et al. (2020)                                                |

<sup>&</sup>lt;sup>1</sup> Production-tagged: Size-resolving through offline lookup tables.

## 2 Methodology

Here we briefly introduce the experimental set-up and ACI relevant components of the 8 GCMs, provide an overview of the 4 remote sensing products used to assess the GCMs, outline the theoretical framework used to disentangle the aerosol effect from meteorological variability, and describe the identification of regions subject to significant SO<sub>4</sub><sup>2-</sup> concentrations attributed primarily to Holuhraun emissions.

<sup>&</sup>lt;sup>2</sup> Refers explicitly to an absence of aerosol-induced precipitation suppression effects on large-scale cloud properties.

<sup>&</sup>lt;sup>3</sup> Aerosol indirect effects simulated in liquid clouds only.

<sup>&</sup>lt;sup>4</sup> With significant developments in the warm rain microphysics by Boutle et al. (2014).

#### 2.1 General Circulation Models

The relevant features of the 8 GCMs that participated in Part 2 of this inter-model comparison study are listed in Table 1. Among the models, 3 types of aerosol modules are employed – modal, sectional, and production-tagged – alongside 5 distinct stratiform cloud microphysics schemes. Of the 8 GCMs, some are based on the same core model. ECHAM6 appears in 3 configurations, each with a different combination of aerosol module and stratiform cloud microphysics scheme, whilst 2 versions of UKESM1 are assessed, one with and one without boundary layer nucleation (BLN). In terms of aerosol activation to cloud droplets, CNRM-ESM2-1 uses the empirical-based parameterisation of Menon et al. (2002), whereas all the other models adopt the Köler theory-based parameterisation of Abdul-Razzak and Ghan (2000). For CNRM-ESM2-1, HadGEM3, and the UKESM1 variants,  $N_d$  is calculated diagnostically, whereas a prognostic approach is taken in the remaining models. In each model, the calculated  $N_d$  from these activation parametrisations is passed to the radiation scheme and used in the calculation of  $r_e$ , which in turn is used to calculate cloud albedo, thereby enabling the simulation of the ACI first indirect effect. The autoconversion parametrisation of Khairoutdinov and Kogan (2000), hereafter KK2000, is adopted in 7 of the GCMs, albeit with different approaches to account for sub-grid variability (see Sect. A). As KK2000 has an inverse power-law dependence on  $N_d$ , an increase in the number of (smaller) cloud droplets acts to inhibit the rate of autoconversion. Hence, the models using KK2000 are able to represent the precipitation suppression mechanism of the ACI second indirect effect. The exception is CNRM-ESM2-1, which instead adopts the autoconversion parameterisation described in Smith (1990) – a scheme not dependent on  $N_d$ , thus preventing aerosol influencing large-scale cloud properties via precipitation suppression. Finally, the additional ACI mechanism of enhanced entrainment evaporation has been shown to be possible in GCMs (e.g., Mülmenstädt et al., 2024), yet the effects are found to be negligible and so will not be considered here.

Performed in their atmosphere-only configurations using prescribed sea surface temperature and sea ice fraction ("AMIP-style"), each model provided simulations with and without Holuhraun  $SO_2$  emissions – the former for 2014 only, whilst the latter for 2002–2014 that acts as a long-term control. To reduce model internal variability and to obtain a model meteorology that closely resembles the observed meteorology during the eruption, horizontal winds are constrained ("nudged") towards ERA-Interim reanalysis (Dee et al., 2011) on a 6-hourly timescale. The Holuhraun simulations distribute the volcanic  $SO_2$  equally between 0.8 and 3 km within the grid cell containing the eruption vent following the emissions profile shown in Table 2 which is based on empirical estimates by Thordarson and Hartley (2015). Both Holuhraun and control simulations contain additional background  $SO_2$  emissions from anthropogenic and natural sources, including passive degassing volcanoes, in accordance with AeroCom Phase III guidelines. Where possible, in-cloud diagnostics directly outputted from the models are used (i.e., model performs necessary calculations during simulation), rather than dividing grid cell mean values by mean CF post-simulation. All model output is regridded to a regular  $1.0^{\circ} \times 1.0^{\circ}$  latitude—longitude grid using linear interpolation, aside from precipitation diagnostics which use first-order conservative interpolation to preserve precipitation totals.

**Table 2.** Sulphur dioxide ( $SO_2$ ) emissions profile used to represent the Holuhraun eruption. Emissions are prescribed in the grid cell containing the eruption vent ( $64.85 \,^{\circ}$ N,  $16.83 \,^{\circ}$ W) and follow empirical estimates by Thordarson and Hartley (2015).

| Days since 31 <sup>st</sup> August | SO <sub>2</sub> emission rate          |  |  |  |
|------------------------------------|----------------------------------------|--|--|--|
| Days since 31 August               | $(kT 	ext{ of } SO_2 	ext{ day}^{-1})$ |  |  |  |
| 0 – 13                             | 100                                    |  |  |  |
| 14 - 30                            | 57.5                                   |  |  |  |
| 31 - 37                            | 80                                     |  |  |  |
| 38 – 91                            | 45                                     |  |  |  |
|                                    |                                        |  |  |  |

## 2.2 Satellite Observations

## 120 2.2.1 MODIS

This study uses the Moderate Resolution Imaging Spectroradiometer (MODIS) MCD06COSP version 6.2.0 Level-3 product (Pincus et al., 2023) to quantify the volcanic impact on cloud properties. The MCD06COSP dataset combines observations from MODIS instruments on-board the Aqua and Terra satellites obtained using the 3.7 µm Cloud Optical Properties Retrieval Algorithm (Platnick et al., 2017). The Level-3 data are outputted at daily and monthly time scales to a regular 1.0° x 1.0° latitude-longitude grid having been sampled from pixel-scale (Level-2) data. This pixel-scale data estimates cloud properties for sunlight pixels (solar zenith angle 

Total effect = 
$$Hol_{14} - NoHol_{clim}$$
. (2)

Note, we remove the year 2014 from NoHolclim to avoid double-counting/dilution of the meteorological variability.

## 2.3.2 Aerosol-only Effect

As the models are nudged, meteorological differences between  $Hol_{14}$  and the 2014 simulations without the eruption (NoHol<sub>14</sub>) are negligible (i.e.,  $\delta m \approx 0$ ). For this special case, Eq. 1 approximates to,

170 
$$\delta c \approx \delta a \frac{\partial c}{\partial a}$$
. (3)

Figure 1. The multi-model ensemble mean perturbation in sulphate  $(SO_4^{2-})$  column load for (a) September and (b) October 2014. Perturbation depicted is the aerosol-only anomaly with meteorological variability excluded (i.e.,  $Hol_{14} - NoHol_{14}$ ) and is expressed in Dobson units (DU). Predominantly volcanically-polluted (PVP) regions are defined over ocean areas where the  $SO_4^{2-}$  column load anomaly exceeds 0.2 DU and anthropogenic aerosol load is low (see main text). These PVP regions are outlined by dotted lines with corresponding spatial mean listed above.

Hence, we estimate the aerosol-only effect on a cloud property using,

Aerosol-only effect = 
$$Hol_{14} - NoHol_{14}$$
. (4)

## 2.3.3 Meteorology-only Effect

With background aerosol largely the same for each year within a particular model, differences in aerosol between NoHol<sub>14</sub> and NoHol<sub>clim</sub> are negligible (i.e.,  $\delta a \approx 0$ ). In this instance, Eq. 1 approximates to,

$$\delta c \approx \delta m \frac{\partial c}{\partial m}.\tag{5}$$

Hence, we estimate the meteorology-only effect on a cloud property using,

$$Meteorology-only effect = NoHol_{14} - NoHol_{clim}.$$
(6)

#### 2.4 Predominantly Volcanically-Polluted Regions

This study focuses on a northern region of the North Atlantic Ocean which is distanced from the anthropogenic aerosol sources of continental Europe. Here low aerosol optical depth values ranging from 0.04 to 0.15 – indicative of "near-pristine to clean" conditions – feature consistently in September–November climatologies derived from a range of datasets (e.g., global aerosol reanalyses – Xian et al. (2024); global atmospheric models – Gliß et al. (2021); Li et al. (2022); and remote sensing instruments – Bevan et al. (2012); Remer et al. (2008)). Clouds in such areas are likely more susceptible to changes in aerosol concentrations making volcanic impacts on ACIs more apparent and easier to isolate.

In the absence of suitable  $SO_4^{2-}$  observations and knowing the models capture the spatial and chemical evolution of the plume with sufficient fidelity (see Part 1, Jordan et al., 2024), we use modelled  $SO_4^{2-}$  column load to identify areas subject to high aerosol loading due to the eruption. As the background aerosol levels in comparison are low, we denote these areas predominantly volcanically-polluted (PVP) regions. We avoid using  $SO_2$  to distinguish PVP regions due to limitations in assuming the co-existence of  $SO_2$  and  $SO_4^{2-}$  including divergent spatial dispersions, time lag in  $SO_2$ -to- $SO_4^{2-}$  conversion, and differing deposition rates. The multi-model ensemble  $SO_4^{2-}$  column load aerosol-only anomaly (i.e.,  $Hol_{14} - NoHol_{14}$ ) for September and October 2014 is shown in Fig. 1. The additional  $SO_2$  emissions from Holuhraun clearly increase the  $SO_4^{2-}$  concentrations within the region; more so in September when the prescribed  $SO_2$  emission rate is higher. The added aerosol loading is not uniformly distributed due to each month's differing meteorological conditions. To identify the PVP regions, we mask the grid cells over land, as well as grid cells with  $SO_4^{2-}$  column load anomalies below 0.2 Dobson Units (DU). The former removes areas likely influenced by anthropogenic pollution, whilst the latter helps ensure a sufficient aerosol concentration to prompt ACI responses.

Nevertheless, despite the isolated nature of the perturbed region, continental pollution may be transported into the area under specific meteorological conditions. Peace et al. (2024) use air mass back-trajectory modelling to show that between the  $15^{th}$ – $21^{st}$  September, anomalous easterly winds brought anthropogenic pollution to the area south of Iceland, mixing with the aerosol load introduced by Holuhraun and hence diluting the volcanic influence there. Malavelle et al. (2017) further show that the anthropogenic pollution is the dominant cause of  $r_e$  perturbations in this area of mixed pollution, meaning any volcanic signal is likely weak. To keep the pollution in the September PVP region  $\sim 100$  % volcanic, we exclude areas below 62° N. Both monthly PVP regions and their associated multi-model ensemble  $SO_4^{2-}$  loading are outlined in Fig. 1. Unless otherwise stated, all values hereafter refer to these PVP regions and not – as is often the case in other Holuhraun studies (e.g., Chen et al., 2022; Malavelle et al., 2017) – the entire domain. Using PVP regions, coupled with the framework laid out in Sect. 2.3, will help attribute any cloud modifications found in this study to volcanic emissions.

#### 3 ACI First Indirect Effect

200

205

The total anomaly (i.e.,  $Hol_{14} - NoHol_{clim}$ ) in cloud top  $r_e$  for September 2014 observed by MODIS is shown in Fig. 2a along-side the associated spatial mean of the PVP region. We evaluate the null hypothesis – that the increase in aerosol concentration had no effect on a cloud property – at each grid cell using a two-tailed Student's t—test. As these "local" hypothesis tests are mutually correlated, to avoid overstating their collective significance we apply the False Discovery Rate (FDR) method (Wilks, 2006, 2016) to control the overall expected proportion of false positives across the spatial domain to 10 %. Stippling highlights grid cells where the null hypothesis is rejected after applying the corrective FDR adjustment (i.e., the cloud property change is unlikely due to random variation). There is a clear decrease in  $r_e$  observed across the northern North Atlantic Ocean, particularly south-east of Iceland where anomalies can exceed -3.00  $\mu$ m. The observed decrease in this area is greater than the PVP region and is likely due to the additional continental anthropogenic aerosol introduced by the meteorological conditions at the time (see Sect. 2.4). The associated modelled total anomalies in cloud top  $r_e$  are shown in Fig. 2b—h. All models capture the ob-

Figure 2. Monthly mean anomalies in cloud droplet effective radius  $(r_e)$  at cloud top for September 2014 from (a) MODIS instruments on-board Aqua and Terra satellites, (b) multi-model ensemble, and (c - h) individual models. Anomalies depicted are the total effect, so include both aerosol and meteorological components (i.e.,  $Hol_{14} - NoHol_{clim}$ ). The predominantly volcanically-polluted (PVP) region is outlined by a dashed line with its spatial mean listed above. Stippling highlights grid cells with null hypothesis rejections based on applying the False Discovery Method (FDR) at a 10 % control level (see main text). Hatched areas indicate missing data. Note that the total effect on  $r_e$  at cloud top cannot be calculated for ECHAM6-HAM-P3 and ECHAM6-SALSA from the output provided to this experiment.

served  $r_e$  anomalies within the PVP region well, notably CNRM-ESM2-1 which agrees to 2 decimal places. The multi-model ensemble mean is within 4 % of the MODIS mean, differing by only  $0.07~{\rm Wm^{-2}}$ . The GCMs do underestimate the observed decrease in  $r_e$  around the UK and Ireland where the transported continental anthropogenic aerosol exists; a discrepancy likely due to differences in the magnitude of background anthropogenic emissions between the real-world and simulated, rather than in the meteorological conditions given that the model runs are nudged. Evidence for a decrease in cloud top  $r_e$  during October is also observed, with the GCMs in good agreement (see Fig. C2). Similar spatial figures for cloud top  $N_d$  can be found in the Appendices (see Figs. B1 and C3).

225

Figure 3. Disentanglement of the aerosol and meteorological effects on (a) cloud droplet number concentration ( $N_d$ ) and (b) cloud droplet effective radius ( $r_e$ ) at cloud top within the predominantly volcanically-polluted (PVP) region for September 2014. Total, aerosol-only, and meteorology-only effects are depicted by green—no pattern, red—minor diagonal, and blue—major diagonal box plots respectively. Box plots extend to the  $25^{th}$ – $75^{th}$  percentiles with outer whiskers at  $5^{th}$ – $95^{th}$ . Black squares depict means. Green bounding and dashed lines extend the observed total effects across rows for visual comparison with the model responses. Climatological baselines are given in brackets. Note that solely the aerosol-only effect can be calculated for ECHAM6-HAM-P3 and ECHAM6-SALSA from the output provided to this experiment.

A comprehensive disentanglement of the aerosol and meteorological effects on cloud top  $N_d$  and  $r_e$  for the PVP regions of September and October 2014 are shown in Fig. 3 and Fig. C3 respectively, with summary values provided in Tables D1 and D2. The box plots are generated from the monthly mean anomalies of the individual grid cells within the PVP region. For example, the September PVP region includes 579 grid cells, meaning 579 "local" values form the dataset used to construct the September box plots. The effect described by a box plot (i.e., total, aerosol-only, or meteorology-only) is dependent on the anomaly the values represent (see Sect. 2.3). MODIS retrievals depict an increase in  $N_d$  which, with the aforementioned observed decrease in  $r_e$ , shows that an ACI first indirect effect initiated by Holuhraun aerosol features in the remote sensing record. The total effect modelled by the individual GCMs all follow the observed directional change for  $N_d$  and  $r_e$ . This, coupled with the component analysis showing that these changes are chiefly aerosol-induced, evidences the ability of the GCMs to capture the ACI first indirect effect within the PVP regions following the eruption, albeit with differing magnitudes. It is worth mentioning that, despite the varying strengths of the model responses, the multi-model ensemble is in good agreement with the observed cloud modifications highlighting the advantages of ensemble based techniques. Note that ECHAM6-HAM-P3 and ECHAM6-SALSA output provided to the experiment make it only possible to calculate the aerosol-only effect on  $N_d$  and  $r_e$ .

The variations in the ACI first indirect effect model representations can largely be explained by their configurations. For example, the strong response in  $N_d$  in ECHAM6-SALSA compared to the other two ECHAM6 models is likely due to the type of aerosol module employed. Sectional schemes, such as HAM-SALSA, better capture small particle growth following a pollution event than modal schemes, such as HAM-M7, due to their ability to resolve finer size distributions and nucleation events, generating more CCN and, subsequently,  $N_d$  (e.g., Matsui and Mahowald, 2017; Mann et al., 2012; Saponaro et al., 2020). For highly-polluted regions, as is the case here, these differences in microphysics can be exacerbated (Kokkola et al., 2018). In addition, the UKESM1 responses with and without BLN imply that including BLN leads to — somewhat counterintuitively — lower  $N_d$  following the introduction of volcanic emissions. The rationale is that the newly nucleated particles from BLN are lofted vertically into the plume where they compete with the aerosol for condensible vapour which hinders the growth of individual particles to CCN size, reducing the number available to form cloud droplets (i.e., clouds in the BLN simulations are less susceptible to increases in aerosol). Finally, despite similar increases in  $N_d$ , HadGEM3 simulates a considerably larger decrease in  $r_e$  than UKESM1 and UKESM1-BLN. This is expected due to improvements added to UKESM1 in aerosol processes, including to the cloud droplet spectral dispersion parameterisation (Mulcahy et al., 2018).

#### 4 ACI Second Indirect Effect

Delaying precipitation formation lies at the heart of the ACI second indirect effect, so it is useful to first assess precipitation totals as even a substantial aerosol perturbation, such as Holuhraun, cannot suppress precipitation in a non-precipitating cloud. Monthly mean surface precipitation rates for September 2014 are depicted in Fig. 4. Observational data from GPCP shows that the PVP region is subject to an average  $2.70 \text{ mm d}^{-1}$ . This value is only  $0.07 \text{ mm d}^{-1}$  less than the 2002-2014 climatological mean, indicating that 2014 is an average year for precipitation, and not anomalously dry. Individual GCM precipitation rates taken from their  $Hol_{14}$  simulations capture the observed spatial pattern and magnitude well; only a minute difference exists of

**Figure 4.** Monthly mean surface precipitation rates for September 2014 from the (a) Global Precipitation Climatology Project (GPCP), (b) multi-model ensemble, and (d - k) individual models, as well as (c) the climatological September mean (2002–2014) derived from GPCP. The predominantly volcanically-polluted (PVP) region is outlined by a dashed line with its spatial mean listed above. Modelled precipitation rates are for the simulations including Holuhraun emission (i.e.,  $Hol_{14}$ ).

Figure 5. Monthly mean anomalies in all-sky liquid water path (LWP) for September 2014 from (a) MODIS instruments on-board Aqua and Terra satellites, (b) multi-model ensemble, and (c - j) individual models. Anomalies depicted are the total effect, so include both aerosol and meteorological components (i.e.,  $Hol_{14} - NoHol_{clim}$ ). The predominantly volcanically-polluted (PVP) region is outlined by a dashed line with its spatial mean listed above. Stippling highlights grid cells with null hypothesis rejections based on applying the False Discovery Method (FDR) at a 10 % control level (see main text).

0.01 mm d<sup>-1</sup> between the multi-model ensemble and GPCP data across the PVP region. Higher precipitation rates are found within the wider domain, notably south-west of Iceland. Observed precipitation in October is slightly higher, yet also average in comparison to the climatology, and is well captured by the GCMs (see Fig. C4). As evidence exists of appreciable precipitation in both the GPCP data and GCMs, there should be scope for the added aerosol from Holuhraun to influence precipitation processes — and subsequently bring forth changes related to the second indirect effect — within both the real-world and modelled cloud systems.

260

We explore the spatial pattern of a possible second indirect effect using LWP – a common proxy for precipitation suppression. The total perturbation in all-sky LWP observed by MODIS during September 2014 is shown in Fig. 5a. As before, stippling indicates grid elements with rejected null hypotheses after applying the FDR method at 10 %. Within the PVP region, our statistical testing identifies a small area where the observed change in LWP is unlikely due to random variation, suggesting the clouds here are retaining more water as a result of the aerosol perturbation introduced by Holuhraun (i.e., precipitation suppression). However, for the vast majority of the domain, any aerosol influence on the observed changes in LWP – both inside and outside the PVP region – are relatively minor compared to meteorological variability. Here, specifically, the positive correlation between LWP and precipitation is shown – areas with higher (lower) cloud liquid water content often support more (less) cloud droplet formation, and so increased (decreased) precipitation. Modelled total anomalies in LWP are depicted in Fig. 5b–j. Whilst the GCMs capture the spatial patterns well, there is clear variation in the magnitude of the anomalies with CAM5.3-Oslo, and the ECHAM6 variants showing strong LWP responses relative to MODIS in and near the PVP region. Nonetheless, the response of the multi-model ensemble differs only slightly to the observed ( $\Delta$ LWP = 4.49 gm<sup>-2</sup>). However, unlike the ACI first indirect effect, the multi-model ensemble response in October (see Fig. C5) is only  $\sim$  50 % that of MODIS, suggesting the September response is likely coincidental. Aside from this, similar observed and modelled behaviour is found for October. Equivalent spatial figures for CF can be found in the Appendices (see Figs. B2 and C6)

A breakdown of the aerosol and meteorological components of the modelled LWP and CF responses alongside MODIS observations for the PVP regions of September and October 2014 is given in Fig. 6 and Fig. C7 respectively, with summary values provided in Tables D1 and D2. Focusing first on the LWP decomposition, the GCMs clearly diverge in the total effect caused by the eruption, with a roughly equal number of models over- and underestimating the impact noted by MODIS. This discrepancy is due mainly to the variation in the simulated aerosol effects, rather than the meteorological effects. For example, in September the mean meteorological component across the individual GCMs varies by 21.68 gm $^{-2}$ , whilst for aerosol this spread is 38.95 gm $^{-2}$  – almost double. Across the two months, the two UKESM1 variants and HadGEM3 simulate a moderate aerosol response ( $\sim 4-8~{\rm gm}^{-2}$ ), whereas a considerably stronger aerosol response ( $\sim 20-40~{\rm gm}^{-2}$ ) is simulated in CAM5.3-Oslo and the three ECHAM6 variants. As expected, the aerosol response for CNRM-ESM2-1 is negligible due to the absence of an aerosol-precipitation mechanism within this model (see Sect. 2.1).

To investigate the moderate and strong aerosol responses in LWP, we explore the aerosol-only effect on the monthly mean rate of cloud droplet autoconversion for September and October in Fig. 7 and Fig. C8 respectively. For the models with autoconversion data available, the volcanic aerosol acts to decrease the rate of autoconversion. This is likely due to the simulated increase in  $N_d$  acting to inhibit cloud droplet growth via the  $N_d^{-1.79}$  inverse dependence in the KK2000 parameterisation that this subset of models all use. Harder to interpret is the variation in the magnitude of the autoconversion decrease. One contributing factor may be how the GCMs represent sub-grid variability in cloud liquid water, which influences the  $q_{cl}^{2.47}$  term in KK2000 that can counteract the suppression driven by  $N_d$ . For instance, implicit descriptions, such as those in the ECHAM6 models, may underestimate this non-linear offset in autoconversion if its scaling term does not accurately reflect the local areas of high cloud liquid water; a feat particularly challenging given the non-uniform conditions of volcanic plumes. Whereas explicit descriptions involving probability density functions, such as the log-normal and Gamma distributions used

**Figure 6.** Disentanglement of the aerosol and meteorological effects on (a) all-sky liquid water path (LWP) and (b) total cloud fraction (CF) within the predominantly volcanically-polluted (PVP) region for September 2014. Total, aerosol-only, and meteorology-only effects are depicted by green—no pattern, red—minor diagonal, and blue—major diagonal box plots respectively. Box plots extend to the 25<sup>th</sup>–75<sup>th</sup> percentiles with outer whiskers at 5<sup>th</sup>–95<sup>th</sup>. Black squares depict means. Green bounding and dashed lines extend the observed total effects across rows for visual comparison with the model responses. Climatological baselines are given in brackets.

in the UKESM1 variants and CAM5.3-Oslo respectively, are potentially able to better capture this non-linear impact on autoconversion from these local high values. Furthermore, it is worth noting that none of the GCMs explicitly account for sub-grid

Figure 7. Monthly mean anomalies in the rate of cloud droplet autoconversion for September 2014 from (a - e) select individual models, and (f) multi-model ensemble. Model responses depict aerosol-only anomalies (i.e.,  $Hol_{14} - NoHol_{14}$ ). The predominantly volcanically-polluted (PVP) region is outlined by a dashed line with its spatial mean listed above. Note that the aerosol-only effect on cloud droplet autoconversion cannot be calculated for HadGEM3 and ECHAM6-HAM from the output provided to this experiment, whilst CNRM-ESM2-1 is not considered here (see main text).

variability in  $N_d$  which, whilst unlikely to contribute to the variation in responses, may cause an underestimation of the suppression via  $N_d^{-1.79}$  if sufficiently high  $N_d$  localities exist.

Regarding the total CF, no substantial overall change is observed by MODIS within the PVP regions of either month — a finding emulated by the models. The aerosol-meteorology decomposition made possible by the GCMs, suggests that the meteorological variability dominates the total effect on CF at the monthly scale, making any conclusion on the aerosol related impact challenging. Nevertheless, a minor increase in total CF due to the added aerosol is simulated by all models except for CAM5.3-Oslo.

#### 5 Top-of-Atmosphere Radiative Response

Here we examine the influence of the volcanic aerosol introduced by Holuhraun on the Earth's energy budget. The total effect on ToA upwelling SW radiation (rsut) for September 2014 given by CERES-EBAF is illustrated in Fig. 8a where increased upward radiative flux is treated as a negative change. Once again, local null hypothesis tests subject to the FDR method at 10 % were conducted. There is mainly an observed increase in rsut across the North Atlantic Ocean following the eruption,

**Figure 8.** Monthly mean anomalies in top-of-atmosphere upwelling shortwave radiation (rsut) for September 2014 from (a) CERES-EBAF, (b) multi-model ensemble, and (c - j) individual models. Anomalies depicted are the total effect, so include both aerosol and meteorological components (i.e.,  $Hol_{14} - NoHol_{clim}$ ). Here radiative fluxes are positive downward. The predominantly volcanically-polluted (PVP) region is outlined by a dashed line with its spatial mean listed above. Stippling highlights grid cells with null hypothesis rejections based on applying the False Discovery Method (FDR) at a 10 % control level (see main text).

with the few areas subject to opposing behaviour largely near land masses in the south (e.g., Celtic Sea, Irish Sea, Baltic Sea, Labrador Sea). Some of the same meteorological features as those depicted in the LWP response are present suggesting again that meteorological variability is clouding any possible observable aerosol signal on rsut. The associated modelled total effects are shown in Fig. 8b – j. The observed spatial pattern is captured well by the models, yet the magnitude varies with most GCMs overestimating the increase in rsut. This discrepancy is most apparent between 45–55 ° N. For October, an improvement in the model performance is noted, with only a difference of 0.09 Wm<sup>-2</sup> between CERES-EBAF and the multi-model ensemble (see

**Table 3.** Radiative forcing (RF) estimates from the Holuhraun eruption across the predominantly volcanically-polluted (PVP) regions and globe. Global RF estimates are scaled from RF estimates of the entire Northern Hemisphere above 50 ° N to exclude noise (see main text).

|                      | Local I | PVP RF     | Global RF            |        |  |
|----------------------|---------|------------|----------------------|--------|--|
| Model name           | (Wr     | $n^{-2}$ ) | $(\mathrm{Wm}^{-2})$ |        |  |
|                      | Sep.    | Oct.       | Sep. — Oct.          | Annual |  |
| CAM5.3-Oslo          | -4.43   | -1.08      | -0.09                | -0.015 |  |
| CNRM-ESM2-1          | -3.87   | -2.51      | -0.04                | -0.006 |  |
| ECHAM6-HAM           | -4.39   | -3.30      | -0.12                | -0.020 |  |
| ECHAM6-HAM-P3        | -5.84   | -2.77      | -0.11                | -0.018 |  |
| ECHAM6-SALSA         | -5.68   | -2.99      | -0.19                | -0.032 |  |
| HadGEM3              | -5.41   | -1.55      | -0.12                | -0.020 |  |
| UKESM1.0             | -4.78   | -1.29      | -0.12                | -0.020 |  |
| UKESM1.0-BLN         | -3.94   | -0.92      | -0.09                | -0.015 |  |
| Multi-model ensemble | -4.79   | -2.05      | -0.11                | -0.018 |  |

Fig. C9). Corresponding spatial figures for ToA upwelling LW radiation (rlut) can be found in the Appendices (see Figs. B3 and C10)

The disentanglement of the aerosol signal from the meteorological variability for rsut and rlut for the PVP regions of September and October 2014 are shown in Fig. 9 and Fig. C9 respectively, with summarising values provided in Tables D1 and D2. All models simulate an overall increase in rsut in the PVP regions as is observed by CERES-EBAF, yet most models overestimate this change, particularly in September, with notable examples including ECHAM6-SALSA and CNRM-ESM2-1 that respectively simulate responses 106% and 75% stronger than observed. The modelled decomposition of the overall increase in rsut shows that the newly introduced aerosol is the predominant cause — likely due to increasing cloud albedo — rather than the meteorological component which often acts to oppose this volcanic influence. In comparison, the aerosol effect on LW radiation leaving the Earth system is minor and dominated by meteorological variability. Nevertheless, for all except UKESM1-BLN, this minor effect is to decrease rlut. This is possibly due to changes in the aerosol direct effect, specifically scattering due to the non-absorbing nature of  $SO_4^{2-}$ , or that low clouds have become thicker due to increased LWP and subsequently trap more upwelling LW radiation. Further analysis with additional diagnostics are needed to confirm this (e.g., using Ghan (2013) methodology). Overall, for both observed and modelled responses, increases in rsut outweigh decreases in rlut, suggesting the Holuhraun eruption prompted a net cooling effect on the Earth's energy budget.

Furthermore, we estimate the strength of this cooling effect using the GCMs. As incoming solar radiation is the same across the  $Hol_{14}$  and  $NoHol_{14}$  simulations, the net change in rsut and rlut between them (i.e., the aerosol-only effect) approximates the RF due to Holuhraun. The local RFs for the September and October PVP regions are listed in Table 3. The model responses vary by  $\sim 2~\rm Wm^{-2}$  for both months/PVP regions, with the ECHAM6 variants and UKESM1-BLN generally simulating the strongest and weakest forcings respectively. Overall the RF is stronger in September when the  $SO_2$  emissions are at their

**Figure 9.** Disentanglement of the aerosol and meteorological effects on top-of-atmosphere upwelling (a) shortwave (rsut) and (b) longwave (rlut) radiation within the predominantly volcanically-polluted (PVP) region for September 2014. Total perturbations, and their aerosol and meteorological components, are depicted by green–no pattern, red–minor diagonal, and blue–major diagonal box plots respectively. Box plots extend to the 25<sup>th</sup>–75<sup>th</sup> percentiles with outer whiskers at 5<sup>th</sup>–95<sup>th</sup>. Black squares depict means. Green bounding and dashed lines extend the observed total effects across rows to aid visual comparison with the model responses. Increased upward radiative flux is treated as a negative change. Climatological baselines are given in brackets.

highest and solar insolation is greater. In addition, we determine global RF estimates to allow comparison of the influence Holuhraun had on the Earth's energy system versus other events. Global values are scaled up from RF estimates of the entire Northern Hemisphere above 50° N and ignore RF contributions outside this area; a choice made to reduce the influence of noise, namely from equatorial regions, as changes in ToA fluxes there are unlikely due to Holuhraun given the spatial evolution of the plume evidenced in Part 1 (Jordan et al., 2024). Averaged across September and October, we find that all models display a global negative forcing in response to the additional aerosol, with our multi-model ensemble estimating a value of -0.11  $\pm$  0.04  ${\rm Wm^{-2}}$  ( $\pm$  1 $\sigma$  of the individual model RFs). The ECHAM6-SALSA global RF is nearly twice that of the ensemble, with the additional forcing potentially due to its consistently strong LWP response across September and October relative to the other models. On the other hand, CNRM-ESM2-1 shows the smallest global RF, roughly a third of the ensemble mean, and is likely due to the exclusion of precipitation suppression induced ACI indirect effects within this model. We assume that the difference across the wider Holuhraun domain of  $\lesssim 20~\%$  between the September-October and annual solar insolation means is small enough to allow us to extrapolate our September-October global RFs to provide proxy annual global RFs. Our multi-model ensemble suggests that, averaged over a year, the added aerosol from Holuhraun caused a forcing of  $-0.018 \pm 0.007~\mathrm{Wm^{-2}}$ . Given that Holuhraun released 3.9 Tg of  $\mathrm{SO}_2$  in our simulations over this period (see Table 2), we estimate a global mean annual RF efficiency for the eruption of  $-0.005 \pm 0.002~\mathrm{Wm^{-2}}$  per Tg of SO<sub>2</sub>. In reality, Holuhraun volcanic activity did not cease in October and continued until February, albeit at a lesser extent, and released an estimated total 9.6–11.8 Tg of SO<sub>2</sub>, hence our annual forcing estimates should be considered as minimums.

## 6 Summary and Conclusion

The continuous degassing of the 2014-15 Holuhraun eruption into the lower troposphere resulted in a persistent source of  $SO_4^{2-}$  pollution across the northern North Atlantic Ocean, providing an opportunistic experiment to assess the representation of ACIs in state-of-the-art GCMs. Here we have presented Part 2 of an AeroCom Phase III inter-model comparison two-part study designed to leverage this opportunity and build on previous works utilising GCMs (Gettelman et al., 2015; Jordan et al., 2024; Malavelle et al., 2017). A simple theoretical framework designed to separate the aerosol and meteorological effects on cloud properties is applied to 8 GCMs across regions identified with minimal non-Holuhraun sources of aerosol pollution during September and October 2014. By comparing the resulting decomposition of the cloud responses to observations from a range of remote sensing instruments, we review the ACI model representations and highlight those that deviate away from the observed behaviour.

Regarding the ACI first indirect effect (i.e., Twomey effect), MODIS observations suggest notable increases and decreases in cloud top  $N_d$  and  $r_e$  respectively across the PVP regions of September and October 2014 when compared to their respective long-term averages. All models correctly capture the direction of these observed changes in cloud top  $N_d$  and  $r_e$ , yet the magnitude of their responses vary. Applying our analysis framework shows that the differences in cloud top  $N_d$  and  $r_e$  relative to their climatological values are almost entirely due to the aerosol added by the eruption rather than interannual variability driven by meteorological influence; a finding in agreement with previous studies (e.g., Chen et al., 2022; Malavelle et al.,

2017). Despite the differences in the strength of the aerosol induced model responses — which are largely explainable by configuration choices — the multi-model ensemble representation of the ACI first indirect effect agrees well with MODIS observations, increasing our confidence in using ensemble based methods to explore this mechanism elsewhere.

For the ACI second indirect effect (i.e., rapid adjustments), we show that in both the real-world and modelled cloud systems precipitation is close to the GPCP climatological average and not anomalously dry following the eruption, meaning aerosol invoked precipitation suppression is possible. We use all-sky LWP and total CF as our proxies to assess whether a delay of precipitation formation is causing macrophysical changes in the clouds. Unlike the microphysical changes in  $N_d$  and  $r_e$ , an aerosol response in LWP and CF is far harder to discern amongst the meteorological variability; a complication previously reported (Malavelle et al., 2017; McCoy and Hartmann, 2015; Peace et al., 2024). Nevertheless, our disentanglement method allows us to isolate the aerosol signal within the PVP regions. The GCMs, excluding CNRM-ESM2-1 for reasons previously stated, show a positive LWP response to the added aerosol suggesting some precipitation suppression is simulated. However, a clear divergence in magnitude is evident, which may stem from differences in implementing the KK2000 autoconversion scheme within the models, particularly in the treatment of sub-grid variability. Given the heterogenous nature of the high aerosol load introduced by Holuhraun, inaccurate representations of sub-grid variability are likely to have a greater impact than under more typical conditions. Moreover, aside from CAM5.3-Oslo, all models simulate a positive volcanic influence on CF, yet the magnitude is minor compared to meteorological variability. In comparison, Chen et al. (2022), via machine-learning techniques, isolate the aerosol signal within MODIS observations and find a far larger increase in CF. If this is the case, then the model CF responses presented here are underestimated and further work to ascertain why is needed.

We show that the volcanic influence on ToA radiation within the PVP regions is predominantly on SW radiation rather than LW with the net effect being an increase in radiation leaving the Earth system. Our multi-model ensemble mean estimates that this cooling has a global radiative forcing of -0.11  $\pm$  0.04 Wm<sup>-2</sup> averaged over September and October, revising previous estimates made using individual GCMs (Gettelman et al., 2015; Malavelle et al., 2017). Such a forcing is comparable to that caused by weak-moderate explosive eruptions (e.g., Kasatochi, Narbo, Sarychev Peak, Raikoke) with SO<sub>2</sub> emissions an order of magnitude less than Holuhraun, yet 10–15 km higher in the atmosphere (Schallock et al., 2023). For Holuhraun, we estimate a global mean annual RF efficiency of -0.005  $\pm$  0.002 Wm<sup>-2</sup> per Tg of SO<sub>2</sub>. For comparison, 2014 global anthropogenic SO<sub>2</sub> emissions had approximately a RF efficiency of -0.010  $\pm$  0.004 Wm<sup>-2</sup> per Tg of SO<sub>2</sub> (Aas et al., 2019; Szopa et al., 2021; Thornhill et al., 2021), whereas a recent reduction in shipping SO<sub>2</sub> emissions incited by 2020 regulations yield a RF efficiency of -0.014  $\pm$  0.002 Wm<sup>-2</sup> per Tg of SO<sub>2</sub> (Jordan and Henry, 2024). Whilst our Holuhraun estimate and these values are in fair agreement, the differences would likely reduce if Holuhraun had occurred during Spring–Summer and/or in a cloud regime more susceptible to aerosol changes as both would act to increase the cooling effect – a notion shared by other studies (Malavelle et al., 2017; Zoëga et al., 2025). Similarly, as the consensus of the GCMs is that the net effect of the meteorological impact acts to oppose the volcanic influence, a greater cooling effect would also occur if Holuhraun had erupted under more favourable meteorological conditions.

Despite best efforts, our study is subject to limitations. Observations are subject to the general constraints of satellite remote sensing at high latitudes, whereas modelling caveats include varied cloud system susceptibility due to differing background

aerosol concentrations across the models, and non-uniformity in the modelled aerosol perturbations/plume representations (e.g., Jordan et al., 2024). Nevertheless, our two-part study of the Holuhraun eruption has used novel techniques to explore 410 GCM representations of ACIs during a high pollution event, confirming their ability to capture the first indirect effect well, whilst highlighting discrepancies in their second indirect effect responses and noting the refinement of their autoconversion schemes as a potential route to improvement.

Code and data availability. The GCM simulation data and code used to produce the results presented here are available at Zenodo via: https://doi.org/10.5281/zenodo.14891975 (Jordan, 2025). All observational datasets used in this study are publicly available. MODIS MCD06COSP version 6.2.0 Level-3 data are available via: https://ladsweb.modaps.eosdis.nasa.gov/ (Pincus et al., 2023). CERES-EBAF ToA Edition 4.2 data are available via: https://ceres.larc.nasa.gov/data/ (Loeb et al., 2018; Kato et al., 2018). GPCP version 3.2 data are available via: https://psl.noaa.gov/data/ (Huffman et al., 2023).

## Appendix A: KK2000 Autoconversion Parameterisation Variations

The majority of GCMs in this study base their autoconversion parametrisation on the scheme by Khairoutdinov and Kogan (2000), hereafter KK2000. KK2000 calculates the rate of autoconversion of cloud water to rain  $\left(\frac{\partial q_{cl}}{\partial t}\Big|_{\text{auto}}\right)$  as a function of cloud liquid water content,  $q_{cl}$ , and cloud droplet number concentration,  $N_d$ . The general form of KK2000 is given by,

$$\left. \frac{\partial q_{cl}}{\partial t} \right|_{\text{auto}} = 1350 \, q_{cl}^{2.47} \, N_d^{-1.79},$$
(A1)

and captures the non-linear dependence of autoconversion on  $q_{cl}$  and  $N_d$ . An important characteristic of KK2000 is that it inhibits the rate of autoconversion following an increase in the number of (smaller) cloud droplets. This enables KK2000 to represent the precipitation suppression mechanism of the ACI second indirect effect (see Sect. 1).

KK2000 was intended for spatial resolutions of  $\mathcal{O}(10 \text{ m})$  – scales able to explicitly resolve cloud-scale processes. To implement KK2000 in large-scale models, such as the GCMs in this study, an additional term is introduced to add a representation of the sub-grid variability of  $q_{cl}$  and/or  $N_d$  – an important factor as the non-linear nature of autoconversion means using grid cell average values will likely add substantial biases (e.g., Larson et al., 2001). The simplest way to account for this sub-grid variability is by applying a scaling term,  $-\gamma_l$ , to Eq. A1 as follows,

$$\left. \frac{\partial q_{cl}}{\partial t} \right|_{\text{auto}} = -\gamma_l 1350 \, q_{cl}^{2.47} \, N_d^{-1.79}. \tag{A2}$$

The tuning of  $\gamma_l$  adjusts the efficiency of the autoconversion rate such that increasing  $\gamma_l$  quickens the rate at which cloud droplets grow by collision-coalescence. Here  $\gamma_l$  implicitly accounts for sub-grid variability. ECHAM6-HAM, ECHAM6-HAM-P3, and ECHAM6-SALSA adopt this form of KK2000 using  $\gamma_l$  values of 10.7, 2.7, and 2.8 respectively. For further information, including the tuning method, see Lohmann and Neubauer (2018) and Dietlicher et al. (2019).

A more detailed approach to characterise sub-grid variability is to use a probability density function (PDF). PDFs can explicitly represent the distribution of  $q_{cl}$  and/or  $N_d$  across a grid cell, capturing those important local features. In CAM5.3-Oslo, a Gamma distribution,  $\Gamma$ , is used to describe the sub-grid variability of  $q_{cl}$ , altering Eq. A1 as follows,

$$\frac{\partial q_{cl}}{\partial t}\bigg|_{\text{auto}} = -CF \frac{\Gamma(\nu + 2.47)}{\Gamma(\nu)\nu^{2.47}} 1350 q_{cl}^{2.47} N_d^{-1.79}, \tag{A3}$$

where CF is cloud fraction and  $\nu$  is a Gamma function shape parameter defined by the relative variance of  $q_{cl}$  within the grid cell. Further details are provided in Morrison and Gettelman (2008) and Gettelman and Morrison (2015).

Moreover, sometimes a log-normal distribution is used to describe the PDF of the the underlying sub-grid variability. This method is done in the configurations of UKESM1 and HadGEM3 assessed in this study. By assuming  $q_{cl}$  follows a log-normal distribution at the sub-grid scale, along with added correction factors, these models implement a refined version of Eq. A1 given as follows,

$$\left. \frac{\partial q_{cl}}{\partial t} \right|_{\text{out}} = E(f_{cl}) \, 1350 \, q_{cl}^{2.47} \, N_d^{-1.79},$$
(A4)

where the sub-grid variability term,  $E(f_{cl})$ , and the parameter  $f_{cl}$  are given by,

$$E(f_{\rm cl}) = (1 + f_{\rm cl}^2)^{-2.47/2} (1 + f_{\rm cl}^2)^{2.47^2/2}$$
(A5)

$$f_{cl} = \begin{cases} (0.45 - 0.25CF)\sqrt{(xCF)^{2/3}} \left( (0.06xCF)^{1.5} + 1 \right)^{-0.17} & \text{if } CF 

Figure B1. Monthly mean anomalies in cloud droplet number concentration  $(N_d)$  at cloud top for September 2014 from (a) MODIS instruments on-board Aqua and Terra satellites, (b) multi-model ensemble, and (c - h) individual models. Anomalies depicted are the total effect, so include both aerosol and meteorological components (i.e.,  $Hol_{14} - NoHol_{clim}$ ). The predominantly volcanically-polluted (PVP) region is outlined by a dashed line with its spatial mean listed above. Stippling highlights grid cells with null hypothesis rejections based on applying the False Discovery Method (FDR) at a 10 % control level (see main text). Hatched areas indicate missing data. Note that the total effect on  $N_d$  at cloud top cannot be calculated for ECHAM6-HAM-P3 and ECHAM6-SALSA from the output provided to this experiment.

Figure B2. Monthly mean anomalies in total cloud fraction (CF) for September 2014 from (a) MODIS instruments on-board Aqua and Terra satellites, (b) multi-model ensemble, and (c - j) individual models. Anomalies depicted are the total effect, so include both aerosol and meteorological components (i.e.,  $Hol_{14} - NoHol_{clim}$ ). The predominantly volcanically-polluted (PVP) region is outlined by a dashed line with its spatial mean listed above. Stippling highlights grid cells with null hypothesis rejections based on applying the False Discovery Method (FDR) at a 10 % control level (see main text).

Figure B3. Monthly mean anomalies in top-of-atmosphere upwelling longwave radiation (rlut) for September 2014 from (a) CERES-EBAF, (b) multi-model ensemble, and (c - j) individual models. Anomalies depicted are the total effect, so include both aerosol and meteorological components (i.e.,  $Hol_{14} - NoHol_{clim}$ ). Here radiative fluxes are positive downward. The predominantly volcanically-polluted (PVP) region is outlined by a dashed line with its spatial mean listed above. Stippling highlights grid cells with null hypothesis rejections based on applying the False Discovery Method (FDR) at a 10 % control level (see main text).

Figure C1. Monthly mean anomalies in cloud droplet number concentration  $(N_d)$  at cloud top for October 2014 from (a) MODIS instruments on-board Aqua and Terra satellites, (b) multi-model ensemble, and (c - h) individual models. Anomalies depicted are the total effect, so include both aerosol and meteorological components (i.e.,  $Hol_{14} - NoHol_{clim}$ ). The predominantly volcanically-polluted (PVP) region is outlined by a dashed line with its spatial mean listed above. Stippling highlights grid cells with null hypothesis rejections based on applying the False Discovery Method (FDR) at a 10 % control level (see main text). Hatched areas indicate missing data. Note that the total effect on  $N_d$  at cloud top cannot be calculated for ECHAM6-HAM-P3 and ECHAM6-SALSA from the output provided to this experiment.

Figure C2. Monthly mean anomalies in cloud droplet effective radius  $(r_e)$  at cloud top for October 2014 from (a) MODIS instruments on-board Aqua and Terra satellites, (b) multi-model ensemble, and (c-h) individual models. Anomalies depicted are the total effect, so include both aerosol and meteorological components (i.e.,  $Hol_{14} - NoHol_{clim}$ ). The predominantly volcanically-polluted (PVP) region is outlined by a dashed line with its spatial mean listed above. Stippling highlights grid cells with null hypothesis rejections based on applying the False Discovery Method (FDR) at a 10 % control level (see main text). Hatched areas indicate missing data. Note that the total effect on  $r_e$  at cloud top cannot be calculated for ECHAM6-HAM-P3 and ECHAM6-SALSA from the output provided to this experiment.

Figure C3. Disentanglement of the aerosol and meteorological effects on (a) cloud droplet number concentration ( $N_d$ ) and (b) cloud droplet effective radius ( $r_e$ ) at cloud top within the predominantly volcanically-polluted (PVP) region for October 2014. Total perturbations, and their aerosol and meteorological components, are depicted by green—no pattern, red—minor diagonal, and blue—major diagonal box plots respectively. Box plots extend to the  $25^{th}$ – $75^{th}$  percentiles with outer whiskers at  $5^{th}$ – $95^{th}$ . Black squares depict means. Green bounding and dashed lines visualise the observed total effects across the model responses. Climatological baselines are given in brackets. Note, only aerosol effect available for ECHAM6-HAM-P3 and ECHAM6-SALSA (see main text).

**Figure C4.** Monthly mean surface precipitation rates for October 2014 from the (a) Global Precipitation Climatology Project (GPCP), (b) multi-model ensemble, and (d – k) individual models, as well as (c) the climatological October mean (2002–2014) derived from GPCP. The predominantly volcanically-polluted (PVP) region is outlined by a dashed line with its spatial mean listed above. Modelled precipitation rates are for the simulations including Holuhraun emission (i.e., Hol<sub>14</sub>).

Figure C5. Monthly mean anomalies in all-sky liquid water path (LWP) for October 2014 from (a) MODIS instruments on-board Aqua and Terra satellites, (b) multi-model ensemble, and (c - j) individual models. Anomalies depicted are the total effect, so include both aerosol and meteorological components (i.e.,  $Hol_{14} - NoHol_{clim}$ ). The predominantly volcanically-polluted (PVP) region is outlined by a dashed line with its spatial mean listed above. Stippling highlights grid cells with null hypothesis rejections based on applying the False Discovery Method (FDR) at a 10 % control level (see main text). Hatched areas indicate missing data.

Figure C6. Monthly mean anomalies in total cloud fraction (CF) for October 2014 from (a) MODIS instruments on-board Aqua and Terra satellites, (b) multi-model ensemble, and (c - j) individual models. Anomalies depicted are the total effect, so include both aerosol and meteorological components (i.e.,  $Hol_{14} - NoHol_{clim}$ ). The predominantly volcanically-polluted (PVP) region is outlined by a dashed line with its spatial mean listed above. Stippling highlights grid cells with null hypothesis rejections based on applying the False Discovery Method (FDR) at a 10 % control level (see main text).

**Figure C7.** Disentanglement of the aerosol and meteorological effects on (a) all-sky liquid water path (LWP) and (b) total cloud fraction (CF) within the predominantly volcanically-polluted (PVP) region for October 2014. Total, aerosol-only, and meteorology-only effects are depicted by green—no pattern, red—minor diagonal, and blue—major diagonal box plots respectively. Box plots extend to the 25<sup>th</sup>–75<sup>th</sup> percentiles with outer whiskers at 5<sup>th</sup>–95<sup>th</sup>. Black squares depict means. Green bounding and dashed lines extend the observed total effects across rows for visual comparison with the model responses. Climatological baselines are given in brackets.

Figure C8. Monthly mean anomalies in the rate of cloud droplet autoconversion for October 2014 from (a - e) select individual models, and (f) multi-model ensemble. Model responses depict aerosol-only anomalies (i.e.,  $Hol_{14} - NoHol_{14}$ ). The predominantly volcanically-polluted (PVP) region is outlined by a dashed line with its spatial mean listed above. Note that the aerosol-only effect on cloud droplet autoconversion cannot be calculated for HadGEM3 and ECHAM6-HAM from the output provided to this experiment, whilst CNRM-ESM2-1 is not considered here (see main text).

**Figure C9.** Monthly mean anomalies in top-of-atmosphere upwelling shortwave radiation (rsut) for October 2014 from (a) CERES-EBAF, (b) multi-model ensemble, and (c - j) individual models. Anomalies depicted are the total effect, so include both aerosol and meteorological components (i.e.,  $Hol_{14} - NoHol_{clim}$ ). Here radiative fluxes are positive downward. The predominantly volcanically-polluted (PVP) region is outlined by a dashed line with its spatial mean listed above. Stippling highlights grid cells with null hypothesis rejections based on applying the False Discovery Method (FDR) at a 10 % control level (see main text).

**Figure C10.** Monthly mean anomalies in top-of-atmosphere upwelling longwave radiation (rlut) for October 2014 from (a) CERES-EBAF, (b) multi-model ensemble, and (c - j) individual models. Anomalies depicted are the total effect, so include both aerosol and meteorological components (i.e.,  $Hol_{14} - NoHol_{clim}$ ). Here radiative fluxes are positive downward. The predominantly volcanically-polluted (PVP) region is outlined by a dashed line with its spatial mean listed above. Stippling highlights grid cells with null hypothesis rejections based on applying the False Discovery Method (FDR) at a 10 % control level (see main text).

**Figure C11.** Disentanglement of the aerosol and meteorological effects on top-of-atmosphere upwelling (a) shortwave (rsut) and (b) longwave (rlut) radiation within the predominantly volcanically-polluted (PVP) region for October 2014. Total perturbations, and their aerosol and meteorological components, are depicted by green–no pattern, red–minor diagonal, and blue–major diagonal box plots respectively. Box plots extend to the 25<sup>th</sup>–75<sup>th</sup> percentiles with outer whiskers at 5<sup>th</sup>–95<sup>th</sup>. Black squares depict means. Green bounding and dashed lines extend the observed total effects across rows to aid visual comparison with the model responses. Increased upward radiative flux is treated as a negative change. Climatological baselines are given in brackets.

## Appendix D: Aerosol-meteorology Disentanglement Summary Tables

**Table D1.** September 2014 aerosol-meteorology disentanglement. Shown are the predominantly volcanically-polluted (PVP) regional means of the total, aerosol-only and meteorology-only effects, as well as a climatological baseline, for cloud top cloud droplet number concentration  $(N_d)$ , cloud top cloud droplet effective radius  $(r_e)$ , all-sky liquid water path (LWP), total cloud fraction (CF), top-of-atmosphere upwelling shortwave radiation (rsut), and top-of-atmosphere upwelling longwave radiation (rlut). Note that for ECHAM6-HAM-P3 and ECHAM6-HAM-SALSA only the aerosol responses in cloud top  $N_d$  and  $r_e$  are available (see main text).

| Model name            | (     | Cloud top I | $V_d (\text{cm}^{-3})$ | )     |       | Cloud to | $p r_e (\mu n)$ | n)    | All-sky LWP (gm <sup>-2</sup> ) |       |       |        |  |
|-----------------------|-------|-------------|------------------------|-------|-------|----------|-----------------|-------|---------------------------------|-------|-------|--------|--|
|                       | Total | Aer.        | Met.                   | Clim. | Total | Aer.     | Met.            | Clim. | Total                           | Aer.  | Met.  | Clim.  |  |
| CAM5.3-Oslo           | 79.15 | 73.95       | 5.19                   | 31.66 | -2.55 | -2.09    | -0.45           | 11.60 | 44.40                           | 31.93 | 12.47 | 87.76  |  |
| CNRM-ESM2-1           | 74.23 | 68.95       | 5.28                   | 98.94 | -1.64 | -1.55    | -0.09           | 10.97 | 2.08                            | 0.16  | 1.92  | 65.03  |  |
| ECHAM6-HAM            | 40.42 | 34.53       | 5.90                   | 42.46 | -1.84 | -1.50    | -0.34           | 11.64 | 20.71                           | 21.24 | -0.53 | 103.15 |  |
| ECHAM6-HAM-P3         | -     | 46.17       | -                      | -     | -     | -0.71    | -               | -     | 38.17                           | 35.82 | 2.35  | 212.35 |  |
| ECHAM6-SALSA          | -     | 104.44      | -                      | -     | -     | -0.50    | -               | -     | 43.47                           | 39.11 | 4.36  | 136.42 |  |
| HadGEM3               | 69.40 | 63.25       | 6.15                   | 52.96 | -2.32 | -1.99    | -0.33           | 9.96  | -2.74                           | 6.47  | -9.21 | 88.37  |  |
| UKESM1                | 83.77 | 84.27       | -0.50                  | 44.38 | -1.30 | -1.45    | 0.14            | 10.76 | 2.46                            | 8.13  | -5.67 | 90.86  |  |
| UKESM1-BLN            | 64.91 | 67.35       | -2.45                  | 76.92 | -0.64 | -0.74    | 0.10            | 9.82  | -2.23                           | 4.93  | -7.17 | 96.87  |  |
| Multi-model           | 68.58 | 67.82       | 3.25                   | 57.89 | -1.71 | -1.31    | -0.16           | 10.79 | 18.29                           | 18.48 | -0.19 | 110.10 |  |
| ensemble              | 00.50 | 07.82       | 3.23                   | 31.07 | -1./1 | -1.31    | -0.10           | 10.79 | 10.29                           | 10.40 | -0.19 | 110.10 |  |
| Observed <sup>1</sup> | 49.89 | -           | -                      | 92.85 | -1.64 | -        | -               | 13.92 | 13.80                           | -     | -     | 113.25 |  |

| Model name            |        | Total ( | CF (1) |       |       | rsut ( | $\mathrm{Wm}^{-2}$ ) |        |       | rlut (Wm <sup>-2</sup> ) |       |         |  |
|-----------------------|--------|---------|--------|-------|-------|--------|----------------------|--------|-------|--------------------------|-------|---------|--|
|                       | Total  | Aer.    | Met.   | Clim. | Total | Aer.   | Met.                 | Clim.  | Total | Aer.                     | Met.  | Clim.   |  |
| CAM5.3-Oslo           | -0.031 | -0.002  | -0.029 | 0.864 | -5.24 | -5.49  | 0.25                 | -86.61 | 0.68  | 1.06                     | -0.39 | -213.28 |  |
| CNRM-ESM2-1           | -0.005 | 0.001   | -0.006 | 0.807 | -5.83 | -3.95  | -1.88                | -81.03 | 1.92  | 0.08                     | 1.85  | -217.23 |  |
| ECHAM6-HAM            | -0.006 | 0.004   | -0.010 | 0.872 | -4.54 | -4.70  | 0.16                 | -80.30 | 0.45  | 0.31                     | 0.14  | -219.17 |  |
| ECHAM6-HAM-P3         | -0.015 | 0.009   | -0.023 | 0.888 | -4.54 | -6.55  | 2.01                 | -84.73 | 0.60  | 0.71                     | -0.11 | -223.25 |  |
| ECHAM6-SALSA          | -0.003 | 0.009   | -0.012 | 0.872 | -6.88 | -6.39  | -0.49                | -79.70 | 0.99  | 0.72                     | 0.27  | -218.43 |  |
| HadGEM3               | -0.039 | 0.002   | -0.041 | 0.914 | -2.73 | -6.07  | 3.34                 | -87.26 | -1.47 | 0.66                     | -2.14 | -217.44 |  |
| UKESM1                | -0.009 | 0.006   | -0.015 | 0.913 | -3.17 | -5.52  | 2.34                 | -85.64 | -1.40 | 0.74                     | -2.13 | -215.37 |  |
| UKESM1-BLN            | -0.014 | 0.002   | -0.017 | 0.918 | -1.09 | -3.80  | 2.71                 | -88.43 | -1.93 | -0.13                    | -1.80 | -214.56 |  |
| Multi-model           | -0.015 | 0.004   | -0.019 | 0.881 | -4.25 | -5.31  | 1.06                 | -84.21 | -0.20 | 0.52                     | -0.54 | -217.34 |  |
| ensemble              | -0.013 | 0.004   | -0.019 | 0.001 | -4.23 | -5.51  | 1.00                 | -04.21 | -0.20 | 0.32                     | -0.54 | -217.54 |  |
| Observed <sup>1</sup> | -0.008 | -       | -      | 0.888 | -3.34 | -      | -                    | -81.27 | 0.20  | -                        | -     | -219.67 |  |

 $<sup>^1</sup>$  MODIS observations used for cloud top  $N_d$ , cloud top  $r_e$ , all-sky LWP, and CF. CERES-EBAF observations used for rsut and rlut.

Table D2. October 2014 aerosol-meteorology disentanglement. Shown are the predominantly volcanically-polluted (PVP) regional means of the total, aerosol-only and meteorology-only effects, as well as a climatological baseline, for cloud top cloud droplet number concentration  $(N_d)$ , cloud top cloud droplet effective radius  $(r_e)$ , all-sky liquid water path (LWP), total cloud fraction (CF), top-of-atmosphere upwelling shortwave radiation (rsut), and top-of-atmosphere upwelling longwave radiation (rlut). Note that for ECHAM6-HAM-P3 and ECHAM6-HAM-SALSA only the aerosol responses in cloud top  $N_d$  and  $r_e$  are available (see main text).

| Model name            | C     | loud top I         | $V_d$ (cm <sup>-3</sup> | <sup>3</sup> )     |       | Cloud to | $p r_e (\mu n$ | n)    | All-sky LWP $(gm^{-2})$ |       |        |               |  |
|-----------------------|-------|--------------------|-------------------------|--------------------|-------|----------|----------------|-------|-------------------------|-------|--------|---------------|--|
|                       | Total | Aer.               | Met.                    | Clim.              | Total | Aer.     | Met.           | Clim. | Total                   | Aer.  | Met.   | Clim.         |  |
| CAM5.3-Oslo           | 57.98 | 53.02              | 4.96                    | 26.46              | -2.21 | -2.17    | -0.04          | 11.80 | 25.23                   | 19.31 | 5.92   | 61.14         |  |
| CNRM-ESM2-1           | 66.77 | 69.91              | -2.75                   | 98.27              | -1.76 | -1.72    | -0.04          | 10.96 | -0.65                   | -0.14 | -0.51  | 61.49         |  |
| ECHAM6-HAM            | 26.16 | 26.67              | -0.14                   | 41.38              | -0.89 | -1.30    | 0.39           | 11.43 | 0.91                    | 18.89 | -17.98 | 94.03         |  |
| ECHAM6-HAM-P3         | -     | 24.26              | -                       | -                  | -     | -0.79    | -              | -     | 8.72                    | 25.82 | -17.10 | 172.54        |  |
| ECHAM6-SALSA          | -     | 81.68              | -                       | -                  | -     | -0.67    | -              | -     | 30.08                   | 37.60 | -7.52  | 104.28        |  |
| HadGEM3               | 28.22 | 25.84              | 2.38                    | 51.17              | -1.14 | -1.07    | -0.07          | 9.66  | 3.27                    | 3.38  | -0.11  | 82.55         |  |
| UKESM1                | 53.17 | 50.55              | 2.62                    | 44.97              | -0.97 | -1.12    | 0.16           | 10.43 | 7.42                    | 4.35  | 3.07   | 93.78         |  |
| UKESM1-BLN            | 40.72 | 36.52              | 4.19                    | 62.31              | -0.56 | -0.69    | 0.12           | 9.88  | 5.67                    | 3.42  | 2.25   | 96.70         |  |
| Multi-model           | 45.57 | 45.98              | 1.84                    | 54.11              | -1.26 | -1.19    | 0.08           | 10.69 | 10.08                   | 14.08 | -4.00  | 95.81         |  |
| ensemble              | TJ.J1 | <del>-</del> 13.70 | 1.04                    | J <del>1</del> .11 | -1.20 | -1.19    | 0.08           | 10.09 | 10.00                   | 17.00 | 00     | <i>93.</i> 01 |  |
| Observed <sup>1</sup> | 42.85 | -                  | -                       | 91.77              | -1.87 | -        | -              | 14.40 | 21.75                   | -     | -      | 131.87        |  |

| Model name            |        | Total ( | CF (1) |       |       | rsut ( | $\mathrm{Wm}^{-2}$ ) |        |       | rlut ( $\mathrm{Wm^{-2}}$ ) |       |         |  |
|-----------------------|--------|---------|--------|-------|-------|--------|----------------------|--------|-------|-----------------------------|-------|---------|--|
|                       | Total  | Aer.    | Met.   | Clim. | Total | Aer.   | Met.                 | Clim.  | Total | Aer.                        | Met.  | Clim.   |  |
| CAM5.3-Oslo           | 0.018  | -0.002  | 0.020  | 0.842 | -3.52 | -2.16  | -1.36                | -51.28 | 3.30  | 1.08                        | 2.21  | -207.17 |  |
| CNRM-ESM2-1           | 0.023  | 0.001   | 0.022  | 0.728 | -3.61 | -2.61  | -1.00                | -48.16 | 1.58  | 0.10                        | 1.49  | -213.39 |  |
| ECHAM6-HAM            | -0.003 | 0.012   | -0.015 | 0.832 | -1.74 | -3.44  | 1.70                 | -47.66 | -1.14 | 0.13                        | -1.27 | -211.37 |  |
| ECHAM6-HAM-P3         | -0.002 | 0.014   | -0.016 | 0.841 | -2.68 | -3.76  | 1.08                 | -49.21 | 0.19  | 0.99                        | -0.80 | -215.15 |  |
| ECHAM6-SALSA          | 0.012  | 0.016   | -0.003 | 0.828 | -3.93 | -3.88  | -0.05                | -46.22 | 0.05  | 0.89                        | -0.84 | -210.99 |  |
| HadGEM3               | 0.009  | 0.004   | 0.005  | 0.892 | -2.48 | -2.03  | -0.45                | -52.86 | 1.33  | 0.48                        | 0.85  | -210.17 |  |
| UKESM1                | 0.012  | 0.002   | 0.011  | 0.906 | -2.94 | -2.02  | -0.92                | -53.97 | 1.86  | 0.73                        | 1.13  | -206.55 |  |
| UKESM1-BLN            | 0.011  | 0.002   | 0.009  | 0.908 | -2.30 | -1.41  | -0.89                | -54.77 | 1.78  | 0.50                        | 1.28  | -206.32 |  |
| Multi-model           | 0.010  | 0.006   | 0.004  | 0.847 | -2.90 | -2.66  | -0.24                | -50.52 | 1.12  | 0.61                        | 0.51  | -210.14 |  |
| ensemble              | 0.010  | 0.000   | 0.004  | 0.047 | -2.90 | -2.00  | -0.24                | -50.52 | 1.12  | 0.01                        | 0.51  | -210.14 |  |
| Observed <sup>1</sup> | 0.020  | -       | -      | 0.886 | -2.99 | -      | -                    | -49.32 | 1.66  | -                           | -     | -213.42 |  |

 $<sup>^1</sup>$  MODIS observations used for cloud top  $N_d$ , cloud top  $r_e$ , all-sky LWP, and CF. CERES-EBAF observations used for rsut and rlut.

Author contributions. GJ, FM, and JH designed the experiment. GJ handled the satellite remote sensing data. GJ, FM, DWP, DN, AL, MM, and PN provided the modelling contributions. GJ, FM, JH, YC, BJ, DP, AP, and ED analysed the ACI responses. GJ prepared the manuscript with contributions from all co-authors.

Competing interests. At least one of the (co-)authors is a member of the editorial board of Atmospheric Chemistry and Physics.

Acknowledgements. The authors would like to express their gratitude to Dr. Andy Jones for providing support in generating the UKESM1 and HadGEM3 contributions to this experiment. The authors would also like to thank Dr. Brice Foucart and Dr. Inger Karset for helping with the CNRM-ESM2-1 and CAM5.3-Oslo submissions respectively. GJ and JH were funded under the European Union's Horizon 2020 Research and Innovation programme under the CONSTRAIN grant agreement 820829. GJ, FM, JH, BJ, and AP are supported by the Met Office Hadley Centre Climate Programme funded by DSIT. JM, YC, and AP would like to acknowledge funding from the NERC ADVANCE grant (NE/S015671/1).

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
