# Peer review of "How well are aerosol-cloud interactions represented in climate models? – Part 2: Isolating the aerosol impact on clouds following the 2014–15 Holuhraun eruption"

_EGUsphere, 2025_

## Author Comment (AC1)

**Author Response**

We would like to thank the reviewers for their time and constructive feedback on the manuscript. Below is a summary of the major changes we made following the two reviews.

Major changes:

- Strengthened the justification in describing the region as "near-pristine"
- Provided additional clarity on the statistical testing approach used
- Provided additional clarity on the data source of the box plot figures
- Corrected the stippling overlay for the individual model response in the spatial plot figures
- Added spatial plots of cloud droplet number concentration, total cloud fraction, and top-of-atmosphere upwelling longwave radiation
- Added extra details on the ACI pathways within the models, particularly the autoconversion parametrisations
- Additional refences added

More detailed responses and changes to individual comments are provided in the latter pages. We have reproduced the reviewers' comments in black text, followed by our response in blue text. Line numbers are based on the revised manuscript. References are provided at the end of the document.

**Review 1 (Anonymous Referee #1)**

This manuscript compares general circulation model simulations of the effects on clouds and radiation from the 2014-15 Holuhraun volcanic eruption in Iceland to satellite observations of clouds and aerosols.

In general the paper is well written, but needs some important clarifications as noted in specific comments below. I have several major concerns.

**Author's Response**

Thank you for your comments. We have addressed these below in the aim of improving the manuscript.

**Review Comment**

1. I'm confused why the authors use monthly means and do not try to actually use daily data, especially from models to get a more process focus. This lessens the utility of the paper, but I think they have the data to look at.

**Author's Response**

Whilst we acknowledge the use of daily data would enable a more thorough process-focused approach, not all model submissions provided output at this temporal resolution to allow this. Given that contributions to the experiment were provided largely on a voluntary basis (i.e., "pro bono"), it is understandable that such data intensive output (i.e., 3 hourly outputs of several 2D and 3D fields) was not provided. Hence, monthly means were chosen for consistency and – as this follows the approach taken by the majority of other Holuhraun studies – allows for an easier comparison with existing literature (e.g., Chen et al., 2022; Malavelle et al., 2017; McCoy and Hartmann, 2015; Zoega et al., 2023, 2025).

Nevertheless, this study still adopts a process-focused approach where possible, such as the newly expanded section on how changes in the autoconversion parameterisations may influence the aerosol-induced response. We hope this satisfies the reviewer's desire for a "more process focus".

**Author's Changes in Manuscript**

Lines 93 – 105, 290 – 303.

Appendix A.

**Review Comment**

2. The author focus on the 'predominantly volcanically polluted (PVP) region using SO2 values. But many of the effects are outside of the PVP regions. Why is that? This might be a case where the PVP region could be better defined if it were done daily rather than monthly given the evolution of emissions and meteorological variability.

I would strongly advise looking at daily data with at least one model to see if it matters for the correlation and attrition, and the volcanically affected or not.

**Author's Response**

Our PVP regions are defined using $SO_4^{2-}$ concentrations rather than $SO_2$ as stated in the main text. We acknowledge that the spatial plots of the total effect (i.e., combined aerosol and meteorological impacts) show signals outside the PVP regions, we attribute these to dominant meteorological variability. Unlike the aerosol perturbation, which is largely confined to the PVP regions, meteorological effects are not spatially constrained, and thus their effects can extend beyond.

We wish to point out that thanks to our framework to separate aerosol and meteorological effects, our study can still extract meaningful aerosol signals even when meteorology dominates. These aerosol-signals are represented by the red box plots with minor diagonals in the disentanglement box plot figures. To further illustrate this separation, we include a spatial figure below of the modelled aerosol-only effect (b – j) on all-sky LWP for September, alongside the observed total effect (a). Clearly, the aerosol-only signal is largely confined within the PVP region with some signal existing in the "mixed pollution" areas around the UK as expected. In contrast, the observed total effect – where meteorological effects are present also – the signals are seen outside. Hence, supporting our opinion that these signals outside the PVP regions are primarily driven by meteorological variability.

[Figure]

Whilst a finer temporal resolution plume mask may help further define the aerosol perturbation, the issue of co-founding meteorology will still exist, and perhaps be worse. For instance, Peace et al. (2024) generated weekly masks of the Holuhraun plume using $SO_2$ satellite retrievals for September to probe ACIs "in-plume vs out-of-plume". They noted strong meteorological effects at the weekly timescale that made the interpretation of the aerosol influence difficult. For this reason, it helps to focus on monthly timescales to smooth out the meteorological "noise" (noise in the sense that it obscures our aerosol signal). Plus, as mentioned previously, we do not have daily/weekly output from all the models to ensure a consistent analysis at these timescales.

**Author's Changes in Manuscript**

N/A.

**Review Comment**

3. More model description is warranted (see specific comments below).

**Author's Response**

We address these specific comments below.

**Author's Changes in Manuscript**

N/A.

**Review Comment**

Page 4, L88: Absent all volcanic emissions? Or just Holuhraun?

**Author's Response**

The long-term control simulations only exclude Holuhraun $SO_2$ emissions; they contain $SO_2$ emissions from other natural activity (including passive degassing volcanoes) and anthropogenic activity in accordance with AeroCom Phase III guidelines. We have amended this sentence to provide more clarity.

**Author's Changes in Manuscript**

Lines 113 – 115.

**Review Comment**

Page 5, L92: Explain how the GCMs do this (or not). Are aerosols and cloud drops prognostic in all the models? Or are cloud drops diagnostic (set as a function of aerosols and activation). I assume CNRM is diagnostic number, but autoconversion (Kessler) does NOT depend on drop number, while it does for Menon and KK? A bit more explanation is warranted.

**Author's Response**

In CNRM-ESM2-1, HadGEM3, and the UKESM1 variants, cloud droplet number concentration ($N_d$) is calculated diagnostically, whereas a prognostic approach is taken in the remaining models. We have added this, along with a more detailed explanation of the ACI pathways present in the models.

**Author's Changes in Manuscript**

Lines 91 – 98.

**Review Comment**

Page 5, L93: Aerosols affect entrainment in the models? How? It thought most GCMs did not include this.

**Author's Response**

Mülmenstädt et al. (2024) explore the possibility of an aerosol-induced enhancement in entrainment evaporation in GCMs in depth. The study finds that "the model produces greater entrainment in response to higher $N_d$ [cloud droplet number] in Sc [stratocumulus] clouds with high-enough $\mathcal{L}$ [liquid water path] to support strong entrainment". The authors state that this acts as evidence that GCMs can technically capture the ACI entrainment mechanism, yet they also provide several caveats including that, globally, a "$N_d$ increase caused by anthropogenic emissions leads, at best, to a very weak decrease in $\mathcal{L}$". After consideration, we believe our wording of this result in our paper is too definitive and has since been relaxed to better represent the conclusions of Mülmenstädt et al. (2024).

**Author's Changes in Manuscript**

Lines 104 – 106.

**Review Comment**

Page 8, L178: so that implies that the area is not really pristine, but can be affected by anthropogenic emissions as well as Holuhraun.

**Author's Response**

Yes, this is correct. Whilst the region is *typically* near-pristine, the area can still be subject to anthropogenic emissions under certain meteorological conditions. We have tweaked our description to "often near-pristine" to acknowledge this. Furthermore, to strengthen our argument that the region is *typically* near-pristine, we include September—November climatological aerosol optical depth values from various sources (e.g., global aerosol reanalyses – Xian et al. (2024); global atmospheric models – Gliß et al. (2021); Li et al. (2022); and remote sensing instruments – Bevan et al. (2012); Remer et al. (2008)). Finally, we expand on how we exclude the continental pollution brought to the region during the 15th—21st September by anomalous easterly winds, citing the back-trajectory analysis of Peace et al., (2024) and the impact analysis on cloud properties by Malavelle et al., (2017).

**Author's Changes in Manuscript**

Lines 180 – 185, 198 – 206.

**Review Comment**

Page 8, L182: but you are still averaging over months. Since you have the meteorology, why not look at daily data? More points, larger gradients. This smooths out the analysis and reduces the chances your differentials are affected by averaging.

**Author's Response**

See previous comments regarding daily data. We have removed "beyond reasonable doubt" due to important counterpoints made by the reviewer.

**Author's Changes in Manuscript**

Line 207.

**Review Comment**

Page 8, L186: what is the 'null hypothesis'? Are you testing significance? Stippled points are significance? Please clarify.

**Author's Response**

Our null hypothesis here – and throughout the study – is that the increased concentration of aerosol has no effect on the cloud property in question. We evaluate this null hypothesis at each grid cell first (i.e., calculate a p-value for each grid cell). As these "local" hypothesis tests are mutually correlated, there is a realistic chance of overstating their collective significance. For example, running 100 **independent** tests at a 5 % significance level, one would expect 5 false positives by chance. However, if the tests are **correlated** – as if often the case with spatial data – more false positives are expected, meaning a result can appear more "significant" than it is.

To account for the correlation within our data, we apply the False Discovery Rate (FDR) method to ensure the collective significance of these individual tests is not misinterpreted. The FDR method is a corrective measure that controls the overall expected fraction of wrongly rejected local null hypotheses by adjusting the p-values of the individual tests. Whilst the FDR method results in more strict significance testing, it does help improve the "signal-to-noise" ratio, reducing the likelihood of overstating results. For further information see Wilks (2006, 2016).

We have amended the manuscript to provide more clarity on the null hypothesis and FDR adjustment applied.

**Author's Changes in Manuscript**

Lines 210 – 215.

**Review Comment**

Page 9, L190: but the PVP region also does NOT have a significant change.

**Author's Response**

Whilst not as much stippling is found in PVP region, some does exist (~15 % of grid cells) indicating that – at least part of - the observed change in $r_e$ is unlikely to have occurred without the increase in aerosol concentration, which, for the PVP region, is largely due to Holuhraun.

**Author's Changes in Manuscript**

N/A.

**Review Comment**

Page 9, L191: again, this argues that region is polluted.

**Author's Response**

We agree that the area south-east of Iceland is polluted due to anthropogenic aerosol being introduced to the region by an anomalous easterly flow from the European continent. However, this

additional pollution does not flow northwards into our September PVP region – our focus – meaning that the perturbation in this area is ~100 % due to volcanic emissions. The ability to isolate areas from these non-Holuhraun sources of "contamination" is the main motivation behind our PVP region methodology.

**Author's Changes in Manuscript**

N/A.

**Review Comment**

Page 9, L192: I would argue they do NOT capture observations well since the largest observed changes are NOT in PVP regions, while the models have largest changes in the PVP regions.

**Author's Response**

After further consideration, we agree that the models struggle to capture the observed changes south-east of Iceland and have amended the manuscript to reflect this. We propose this discrepancy is due to differences in the magnitude of background anthropogenic emissions between the real-world and simulated. We stand by our statement that within the PVP region – where the aerosol perturbation is ~100 % volcanic – the models do capture the observed behaviour well.

**Author's Changes in Manuscript**

Lines 216 – 219..

**Review Comment**

Page 10, Figure 3: are these box plots individual location averages? What gives the spread. It has not been well defined.

**Author's Response**

The box plots are generated using the monthly mean anomalies from individual grids cells within the PVP region. For example, the September PVP region includes 579 grid cells, meaning 579 "local" values form the dataset used to construct the box plots. The effect described by a box plot (i.e., total, aerosol-only, or meteorology-only) is dependent on the anomaly the values represent (see Sec. 2.3).

We have amended the manuscript to provide more clarity on the data used to generate the box plots.

**Author's Changes in Manuscript**

Lines 228 – 231.

**Review Comment**

Page 11, L208: what about the 'polluted' region S. Of the PVP region? Shouldn't you comment on that: seems like MODIS might have a larger effect than the models.

**Author's Response**

We comment on this discrepancy in the previous paragraph when discussing the spatial $r_e$ figure. As this section (i.e., discussion of the box plots) is purely focused on the PVP regions, we will not add another comment about the discrepancy between MODIS and the models in the wider domain.

**Author's Changes in Manuscript**

N/A.

**Review Comment**

Page 11, L214: exasperated is not the right word. I think you mean 'increased' or exacerbated. That's still a bit awkward.

**Author's Response**

Yes, we meant exacerbated. Thank you for the correction.

**Author's Changes in Manuscript**

Line 244.

**Review Comment**

Page 11, L228: Isn't the frequency of precipitation (and susceptible cloud) also important? It's not really the mean, it's the number of days that it is effective at precip suppression.

**Author's Response**

Agreed, the frequency of days with precipitation is also a factor. However, as we do not have daily data for all models, we cannot calculate this quantity consistently and so opt to focus on mean precipitation only as was done in Malavelle et al. (2017).

**Author's Changes in Manuscript**

N/A.

**Review Comment**

Page 12, L:230: Figure 4: is this a lot or a little precip in the PVP regions. It's hard to tell from the figure or from what you have said here.

**Author's Response**

The observed monthly mean September and October precipitation rates of the PVP regions are within 2.59 % and 0.89 % of their climatological means (2002 – 2014) respectively. This shows that 2014 is an average year for precipitation, and not anomalously dry. We have added a climatological subplot to the precipitation figures to provide this added context.

**Author's Changes in Manuscript**

Lines 256 – 257.

Figures 4 and C4.

**Review Comment**

Page 12, L237: Are they really 'excellently' capturing the spatial distribution of LWP. What are the potential issues in measuring LWP?

**Author's Response**

After further consideration, we agree that "excellently" is overstating the results, particularly given the ECHAM6 configurations appear to miss the positive anomaly in the southern part of the domain. We now relax this statement to "well". A good discussion of the potential issues in measuring LWP are provided in Greenwald et al., 2007.

**Author's Changes in Manuscript**

Line 274.

**Review Comment**

Page 13, L239: I don't think the ensemble matching the magnitude of the observed really means anything: if you only included one model from each family you might get a different answer. Some models have very different patterns. I think you could do more to look at the spatial distribution of effects: why are there large effects OUTSIDE of the PVP regions in models and obs? Do they correlate with aerosols? Nd? Precip even? Does this hold at a daily scale? Not just s smeared out monthly average.

**Author's Response**

We agree that including only a single configuration of each core model would potentially result in a wider discrepancy between the multi-model ensemble and observations. Given that only the September multi-model response in the cloud liquid water path (LWP) anomaly is in good agreement with MODIS and not October, we have removed the statement on the "excellent" ensemble response for the ACI second indirect effect as it is likely just coincidental. However, we do believe there is enough evidence to suggest the ensemble approach for the ACI first indirect effect is beneficial and keep the statements regarding this.

The large effects observed in LWP – both inside and outside the PVP region – are likely due to meteorological variability dominating the aerosol signal. Specifically, here it is likely due to the positive correlation between LWP and precipitation -- areas with higher (lower) cloud liquid water content often support more (less) cloud droplet formation, and so increased (decreased) precipitation. In areas with precipitation well-above or below the average, meteorology is likely the primary driver of the LWP response, such as the area south-west of Iceland. We are unable to comment on the daily scale as this data is not available from all models.

**Author's Changes in Manuscript**

Lines 270 – 279.

**Review Comment**

Page 14, L246: But figure 6 shows some large spread in met effects (large blue box and whiskers) and of different sign. What is going on?

**Author's Response**

The large spread in the meteorology-only effect noted in some models is due to the heterogeneous nature of precipitation. Without the influence of aerosol on LWP, the LWP response will be positively correlated with precipitation (see previous comment). Given the size of the area, it's unlikely that precipitation will be uniform, and so neither will the LWP response, hence the large spread in the meteorology-only effect. Grid cells with above (below) average precipitation will likely have positive (negative) LWP anomalies.

**Author's Changes in Manuscript**

N/A.

**Review Comment**

Page 15, L254: that seems pretty self evident and consistent with lots of other work.

**Author's Response**

Agreed, yet the intention here to introduce the autoconversion parameterisations, rather than express a result. As the statement does this well, we keep it, yet no longer use "interestingly".

**Author's Changes in Manuscript**

Lines 291 – 292.

**Review Comment**

Page 15, L257: you should figure out how KK varies across this subset of models: is it tuned differently?

**Author's Response**

Indeed, the Khairoutdinov and Kogan parameterisation (KK2000) is adopted differently in this subset of models, namely in how sub-grid variability of cloud liquid water content is represented. We have added a section in the Appendices that lists the exact forms of KK2000 in the models. Plus, we have expanded our discussion on this parameterisation within the main text.

**Author's Changes in Manuscript**

Lines 290 – 303.

Appendix A.

**Review Comment**

Page 15, L258: is the lack of CF response shown anywhere?

**Author's Response**

The mean cloud fraction response within the PVP region is shown in Fig. 6b. To provide additional detail, we now include the spatial cloud fraction total anomaly for September and October in the Appendices.

**Author's Changes in Manuscript**

Figures B2 and C6.

**Review Comment**

Page 17, L266: Is a reader to read in figure 8 that almost all of the points have significant differences? even ones where the difference is nearly zero (e.g. regions where it switches from positive to negative). That does not seem correct for significance testing…

**Author's Response**

No, the absence of stippling indicates areas where there is insufficient evidence to reject the null hypothesis (i.e., where increased aerosol concentrations did not lead to a significant change in top-of-atmosphere radiation). As mentioned above, we have expanded on our statistical testing in the main text to add needed clarity.

**Author's Changes in Manuscript**

Lines 210 – 215.

**Review Comment**

Page 17, L268: it's not noise if it is significant? It's significant meteorological variability right?

**Author's Response**

The meteorological variability is noise in the sense in that it is unwanted and obscures the aerosol signal we wish to isolate. However, we understand your point and have removed the reference to "noise".

**Author's Changes in Manuscript**

Line 306.

**Review Comment**

Page 17, L285: if clouds get thicker, then low clouds will trap more LW and reduce OLR. I think it's a consequence of higher TAU (more LWP). The LW offsets the SW somewhat.

**Author's Response**

Thank you, this possible cause has now been included.

**Author's Changes in Manuscript**

Lines 330 – 332.

**Review Comment**

Page 17, L291: also due to more daylight….

**Author's Response**

Thank you, this point has been added.

**Author's Changes in Manuscript**

Line 340.

**Review Comment**

Page 19, L304: this global efficiency seems very dependent on location and timing of emissions no? Is it really relevant? You only put in emissions for Sept and Oct? Did you calculate effects through February? If not, why not? Seems simple to do.

**Author's Response**

While the Holuhraun $SO_2$ emissions extended through to February, there is minimal SW radiation in our PVP regions after October. This, coupled with a substantial reduction in the emissions since the initial eruption, would result in very uncertain estimates of the forcing effects in these later months. Plus, only a couple of model submissions provided data into 2015 – understandably given the "pro bono" nature of this work – so we opted to keep the calculations to September and October for consistency, in line with the rest of our study.

Regarding, the global radiative forcing efficiency, we agree that it is dependent on both the location and timing of the $SO_2$ emissions – a dependency already explored in Malavelle et al. (2017) using a similar version of the UKESM1 model used in our study. We believe our estimate adds another data point for the radiative forcing efficiency of $SO_2$ and helps put Holuhraun in context with the wider literature.

**Author's Changes in Manuscript**

N/A.

**Review 2 (Anonymous Referee #2)**

In this study, the authors utilize the 2014-15 Holuhraun eruption as a natural laboratory through which they explore aerosol-cloud interactions (ACI) in an ensemble of state-of-the-art general circulation models (GCMs). Following on from Part 1 of this paper, which addresses the modeling of the eruption's emissions from the volcanic plume, the authors focus on ACI in marine stratocumulus clouds. To this end, 8 models are assembled with a variety of different cloud microphysics parameterization schemes, with a primary focus on modeled precipitation suppression. The authors contrast the output from models where the eruption is present to models where it isn't and then compare these results to MODIS observations. They find that, while the models seem to adequately capture the Twomey effect (the first indirect effect), there remains considerable disagreement in modeled adjustments (the second indirect effect).

The paper's writing is clean and generally easy-to-follow, and it's well-written overall. However, as it stands, the paper feels lacking in certain details, and some revisions and additions are recommended before publishing. These are described point-by-point below.

**Author's Response**

Thank you for your comments. We have addressed these below in the aim of improving the manuscript.

**Review Comment**

1. In the paper, the authors identify precipitation suppression as the expected mechanism for enhanced adjustments. For the majority of models, the autoconversion parameterization used is Khairoutdinov and Kogan (2000) (hereafter KK2000), which relies on several parameters (Nd exponent etc.) to determine the rate of autoconversion. It is suggested (P20, L333) that differences in the configuration of KK2000 between models may account for these differences. I think this is likely true, but more analysis here to interpret the causality of these differences would make the result more robust. How do these KK2000 parameters compare between models? Do the autoconversion parameters match up with the outcomes shown in Figure 7? Exploring these parameterization differences would be a straightforward pathway towards discerning the causality behind the differences in the model and strengthening the results, allowing the authors to more specifically describe the discrepancies that led to this result.

**Author's Response**

Indeed, the Khairoutdinov and Kogan parameterisation (KK2000) is adopted differently across the models, namely in how sub-grid variability of cloud liquid water content is represented. We have added a section in the Appendices that lists the exact forms of KK2000 in the models. Plus, we have expanded our discussion on this parameterisation within the main text.

**Author's Changes in Manuscript**

Lines 290 – 303.

Appendix A.

**Review Comment**

2. Following on to the previous point, additional explanation of the ACI pathways present in the model would be useful for the interpretation of results. As discussed, precipitation suppression is the main pathway the authors are concerned with in this paper (although, as mentioned, ACI effects on entrainment are present in models, though weakly). How, mechanistically, through the parameterizations implemented, do the authors expect GCM ACI to manifest from the enhanced SO2? Are there significant structural differences between microphysics schemes that may play a part in output disagreement, or are you more concerned with parametric differences?

**Author's Response**

We have expanded the description of the GCMs in the text to include this additional explanation of the ACI pathways within each model. We detail how, mechanistically, the increase in aerosol should alter precipitation via autoconversion.

**Author's Changes in Manuscript**

Lines 93 – 106.

**Review Comment**

3. For this work, only two months are analyzed: "September and October 2014 when the eruption is strongest" (P3, L76). However, this is not the whole eruption, and considerable additional SO2 was emitted after the end of October. Why not continue this analysis through the remaining months, especially given that you've already created the framework? The exclusion of the remaining eruption is not well-motivated, and as-is undermines the completeness of your conclusions. Before publication this should be addressed, either in the form of a stronger motivation for neglecting the remainder of the eruption or by including those months in the analysis.

**Author's Response**

Our study focuses exclusively on September and October as these months offer the most favourable conditions for isolating the aerosol signal from the eruption. Compared to the later months:

1. Satellite retrievals are more reliable as high-latitudes coverage deteriorates during the Northern Hemisphere winter, complicating comparisons with observations
2. $SO_2$ emissions in September and October are at their peak
3. The volcanic plume is well defined with plume dilution at its lowest
4. A plethora of studies have been published providing insights on the conditions for this period

We agree that additional months would benefit our analysis, yet this would only be true if those months offered comparable quality. Given the above, this is not the case and justifies our decision to focus solely on September and October.

**Author's Changes in Manuscript**

Lines 77 – 80.

**Review Comment**

P3, L52: Here and elsewhere, you describe the region as "near-pristine". While I don't disagree that the region can get pristine conditions, I imagine that this must be highly dependent on the prevailing

winds/meteorology (as has been observed in the Azores to the south, see Gallo et al., 2023). To that point, you later describe how anthropogenic emissions from the UK affect a large region below the PVP domain. I don't think this is necessarily a huge problem, given that you're accommodating for anthropogenic emissions by mirroring them in your Hol and NoHol, but if you wish to characterize the region as such, a relevant climatology citation may be useful here. Alternatively, additional specific commentary regarding the prevailing background aerosol regime in this region to the Holuhraun-dominated regime may be useful for "setting the stage".

**Author's Response**

Good point. We have tweaked our description to "often near-pristine" instead to acknowledge that anthropogenic emissions can be introduced to the region under certain meteorological conditions. Furthermore, to strengthen our argument that the region is *typically* near-pristine, we include September—November climatological aerosol optical depth values from various sources (e.g., global aerosol reanalyses – Xian et al. (2024); global atmospheric models – Gliß et al. (2021); Li et al. (2022); and remote sensing instruments – Bevan et al. (2012); Remer et al. (2008)). Finally, we expand on how we exclude the continental pollution brought to the region during the 15th—21st September by anomalous easterly winds, citing the back-trajectory analysis of Peace et al., (2024) and the impact analysis on cloud properties by Malavelle et al., (2017).

**Author's Changes in Manuscript**

Lines 180 – 185, 198 – 206.

**Review Comment**

P8, L186-7: The statistical methods used throughout are a bit unclear. What is the "local" null hypothesis that is being rejected? Can you describe, briefly, the FDR method? More elaboration is required here, especially given how frequently these methods are used throughout the rest of the paper.

**Author's Response**

Our null hypothesis here – and throughout the study – is that the increased concentration of aerosol has no effect on the cloud property in question. We evaluate this null hypothesis at each grid cell first (i.e., calculate a p-value for each grid cell). As these "local" hypothesis tests are mutually correlated, there is a realistic chance of overstating their collective significance. For example, running 100 **independent** tests at a 5 % significance level, one would expect 5 false positives by chance. However, if the tests are **correlated** – as if often the case with spatial data – more false positives are expected, meaning a result can appear more "significant" than it is.

To account for the correlation within our data, we apply the False Discovery Rate (FDR) method to ensure the collective significance of these individual tests is not misinterpreted. The FDR method is a corrective measure that controls the overall expected fraction of wrongly rejected local null hypotheses by adjusting the p-values of the individual tests. Whilst the FDR method results in more strict significance testing, it does help improve the "signal-to-noise" ratio, reducing the likelihood of overstating results. For further information see Wilks (2006, 2016).

We have amended the manuscript to provide more clarity on the null hypothesis and FDR adjustment applied.

**Author's Changes in Manuscript**

Lines 210 – 215.

**Review Comment**

P11, L214: I am unsure if "exasperated" is the right word here- did you mean "exacerbated"?

**Author's Response**

Yes, thank you for the correction.

**Author's Changes in Manuscript**

Line 244.

**Review Comment**

P17, L290-1: While it is not unreasonable that the time during which the eruption was most intense had the highest RF, could this not also have been affected by the differing solar radiation between the two periods of time? At these extreme latitudes, the amount of sunlight changes significantly over the course of September and October – I wonder if that may be influencing the relative RF magnitudes during these months, and if that sunlight difference can be easily normalized between the two periods.

**Author's Response**

Thank you for providing this alternative explanation. We agree that solar insolation would also affect the radiative forcing between the two months, and we now include this point in the text. Regards to normalising the differences in incoming solar radiation (rsdt), unfortunately not all the models provided this variable. However, given how similar rsdt will be across the models, we have used the values from UKESM1.0 to compare the September-October mean rsdt relative to their annual mean across the domain. Whilst there is a difference between the two periods – ~20 % – we feel it is small enough to justify extrapolating the September-October mean radiative forcing to get an annual equivalent as has been done in Malavelle et al. (2017). However, we now add extra emphasis in the manuscript that these estimates should be taken as proxies given the significant assumptions made.

**Author's Changes in Manuscript**

Lines 339 – 340, 349 – 351.

**References**

[revised manuscript text omitted]